# VisDiff: SDF-Guided Polygon Generation for Visibility Reconstruction, Characterization and Recognition

## Abstract

The capability to learn latent representations plays a key role in the effectiveness of recent machine learning methods. An active frontier in representation learning is understanding representations for combinatorial structures which may not admit well-behaved local neighborhoods or distance functions. For example, for polygons, slightly perturbing vertex locations might lead to significant changes in their combinatorial structure (expressed as their triangulation or visibility graph) and may even lead to invalid polygons. In this paper, we investigate representations to capture the underlying combinatorial structures of polygons. Specifically, we study the open problem of *Visibility Reconstruction*: Given a visibility graph $G$, construct a polygon $P$ whose visibility graph is $G$. *Visibility Reconstruction* belongs to the Existential Theory of Reals ($\exists$R) complexity class (which lies between NP and P-SPACE). Currently, reconstruction algorithms are available only for specific polygon classes. Establishing the hardness of the general problem is open.

We introduce **VisDiff**, a novel diffusion-based approach to reconstruct a polygon from its given visibility graph $G$. Our method first estimates the signed distance function (SDF) of $P$ from $G$. Afterwards, it extracts ordered vertex locations that have the pairwise visibility relationship given by the edges of $G$. Our main insight is that going through the SDF significantly improves learning for reconstruction. In order to train VisDiff, we make two main contributions: (1) We design novel loss components for computing the visibility in a differentiable manner and (2) create a carefully curated dataset. We use this dataset to benchmark our method and achieve 21% improvement in F1-Score over standard methods. We also demonstrate effective generalization to out-of-distribution polygon types and show that learning a generative model allows us to sample the set of polygons with a given visibility graph. Finally, we extend our method to the related combinatorial problem of reconstruction from a triangulation. We achieve 95% classification accuracy of triangulation edges and a 4% improvement in Chamfer distance compared to current architectures. Lastly, we provide preliminary results on the harder visibility graph recognition problem in which the input $G$ is not guaranteed to be a visibility graph.

## 1 Introduction

Many types of objects ranging from molecules to organs to maps can be represented geometrically. Polygons are one of the most commonly used geometric representations. They are planar objects specified as a cyclically ordered set of points. The line segments connecting these pairs of points in the given order represent the boundary of an object such as the hand shown in Figure 1-left. As one considers the hands of various people, they realize that shape parameters such as the relative length and thickness of fingers or palm sizes vary across samples. At the same time, intuitively, most hands seem to share a common structure. This intuition can be formalized by studying the underlying combinatorial structures of the corresponding polygons representing the hands. For example, one can triangulate each polygon and construct its dual. The dual, with the appropriate embedding, closely resembles a skeleton (Figure 1-middle). Graphical structures such as the triangulation dual

or the visibility graphs of polygons provide insights about the underlying combinatorial structures of shapes. In this paper, we study representations that link polygons to their combinatorial structures. We study polygons which are simple (the boundary does not self intersect) and simply-connected (no holes).

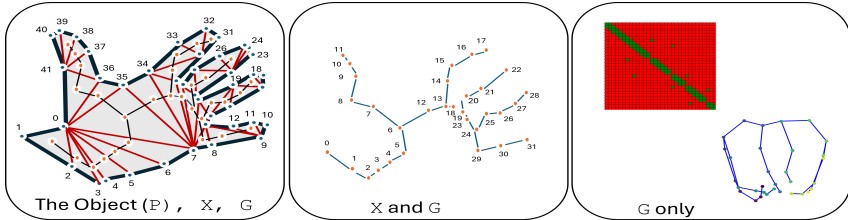

Figure 1: **Left:** An object (a hand) represented as a polygon $P$. The polygon is given by an ordered list of vertex locations $X$. Also shown is a triangulation of $P$ and its dual graph $G$. **Middle:** The dual of the triangulation of $P$. It is represented as a graph $G$ which has a vertex for each triangle and an edge between two adjacent triangles. This drawing contains information about $G$ as well as $X$ because locations from the left figure were used for embedding the graph on the plane. **Right:** $G$ represented as an adjacency matrix generated using edges of the dual by fixing the ordering of the triangles. We seek to answer the question: How much information about $X$ can be recovered from $G$ alone? Also shown in the figure is a standard embedding of $G$ (with Isomap). Clearly, standard graph embedding algorithms are not sufficient to recover $X$ from $G$.

The main question we study is the following: Suppose we are given a graph $G$ representing the combinatorial structure of a polygon. $G$ could be the visibility graph or a triangulation of the polygon. Note that $G$ does not contain any coordinate information $X$. What can we say about the polygon, or the set of polygons, that have this structure $G$? It might be tempting to use standard metric embedding methods such as Isomap (Tenenbaum et al., 2000) to reconstruct $X$ but since $G$ does not admit a natural distance metric such methods are doomed to fail as shown by the example in the right figure. Formally, let $X(P)$ be the vertex locations of a polygon $P$ and $G(P)$ be a graphical property of $P$. In this paper, we consider visibility graphs and triangulations. We consider the following problems in increasing difficulty:

**Problem 1 (Reconstruction)** *Given a valid $G$, generate **a** polygon $P$ such that $G(P) = G$.*

**Problem 2 (Characterization)** *Given a valid $G$, generate **all** polygons $P$ such that $G(P) = G$.*

Note that in these two problems, the input $G$ is assumed to be valid – i.e., there exists a polygon $P$ whose visibility graph or triangulation dual is $G$. While we primarily focus on reconstruction and characterization problems in this paper, we also provide insights into the more general recognition problem in which $G$ is arbitrary:

**Problem 3 (Recognition)** *Given an arbitrary graph $G$, determine whether there exists a polygon $P$ such that $G(P) = G$.*

The primary combinatorial structure we study in this paper is the visibility graph. The visibility graph of $P$, denoted $Vis(P)$ is a graph which has a vertex for each vertex of $P$. There is an edge between two vertices $u$ and $v$ if and only if $u$ and $v$ are visible to each other in $P$. In other words, the line segment connecting them is completely inside $P$. The visibility graph is an important combinatorial structure because it is unique for a given polygon, and contains many other important structures such as triangulations and shortest path trees (Guibas et al., 1986).

**Our contributions:** We present **VisDiff**: a generative model which takes a visibility graph $G$ as input and a seed for diffusion, and first generates a polygon $P$ represented as a signed distance function (SDF). Next, vertex locations on the zero level set are selected so that $Vis(P) = G$. Our main insight is that going through the SDF as an intermediate representation yields superior results over using established methods to predict the vertex locations directly. In order to train VisDiff we design novel loss functions for evaluating the validity of the output polygon and comparing its visibility graph to the input *in a differentiable manner*. We also design a carefully curated dataset which

captures a wide range of combinatorial properties of polygons. Current random polygon generation methods struggle to faithfully represent the visibility graph space. They are biased towards high concavity as the number of points increases. We address this problem by systematically rebalancing the dataset by the link diameter – which quantifies concavity.

We show that VisDiff can also be used for characterization since we can sample the set of polygons which have a given visibility graph. To show the generality of VisDiff, we apply it for the problem of reconstructing a polygon from its triangulation graph. Finally, we present preliminary results on how VisDiff can be used for recognition by turning it into a classification problem based on the difference between the input graph (which may not be a visibility graph) and the visibility graph of the output polygon. This last result suggests that VisDiff is learning a meaningful representation over the space of all polygons. Overall, our results provide evidence that recent architectures can learn representations of non-trivial combinatorial structures such as polygons. We start with overview of related work.

## 2   RELATED WORK

We summarize the related work in three main directions: Visibility graph reconstruction and recognition, representation learning for shapes and graph neural networks.

**Visibility graph reconstruction and recognition**: The problem of reconstructing and recognizing visibility graphs is studied extensively in the computational geometry literature. Yet, it is still an open problem (Ghosh & Goswami, 2013). In the current literature, there are reconstruction and recognition for polygons of certain categories: Ameer et al. (2022) solved the recognition and reconstruction problems for pseudo polygons. Silva (2020) showed that visibility graphs of convex fans are equivalent to visibility graphs of terrain polygons with an addition of a universal vertex. Everett & Corneil (1990) proposed an algorithm to solve the recognition problem in spiral polygons. Boomari & Zarei (2016) proposed reconstruction and recognition algorithm for anchor polygons. Colley et al. (1997) proposed a linear time algorithm to recognize visibility graphs for tower polygon. Dehghani & Morady (2009) solved the reconstruction problem for embedded planar graphs. On the hardness side, the complexity of the visibility graph recognition and reconstruction problem is known to belong to PSPACE (Everett, 1990) specifically in the Existential Theory of the Reals class (Boomari et al., 2018). The exact hardness of the problem is still open. In this work we explore it from the representation learning perspective to understand if generative models can learn the underlying manifold of the space of polygons and their visibility graphs in a generalizable fashion.

**Representation Learning**: 3D shape completion (Chou et al., 2023) (Chen et al., 2024) (Cheng et al., 2023) (Shim et al., 2023) is a closely related application. In 3D shape completion, the input contains partial geometric information for example as a point cloud. In our case, the input is only a combinatorial description such as the visibility graph. There might be many shapes consistent with the input graph and extracting them without any geometric information as part of the input is challenging. Another body of work related to our problem is mesh generation (Gupta et al., 2023) (Wang et al., 2020). Two recent results in this domain are MeshGPT (Siddiqui et al., 2024) and PolyDiff (Alliegro et al., 2023). Both of these approaches generate high-quality 3D triangular meshes by learning to output a set of triangles from a fixed set of triangles. PolyDiff discretizes the 3D space into bins and MeshGPT works over a predefined set of triangles. In our work, we seek to learn the space of all polygons and their visibility graphs.

**Graph Neural Networks (GNNs)**: GNNs are one of the standard representations for graphs. The current literature on GNNs primarily focuses on graphs with features associated with a well-defined metric space. In the literature, the closest to our work is generating graph embeddings for a given distance matrix. Li et al. (2024) showed that a GNN given all-pairwise Euclidean distance information which is known as Vanilla DisGNN, fails to differentiate between symmetric graph structures. To address the limitation of Vanilla DisGNN, they propose $k$-DisGNN. $k$-DisGNN captures information not just from immediate neighbors but from a $k$-hop neighborhood around each node. The ability to utilize the k-hop neighbourhood results in building richer geometric representations for differentiating between symmetric structures efficiently. Cui & Wei (2023) proposed MetricGNN, which is capable of generating graph embedding from a given embedding distance matrix. Shi et al. (2021) proposed ConfGF which uses GNN for determining molecular conformation given the inter-atomic distances and bond characteristics. Yu et al. (2024) proposed a GNN architecture, PolygonGNN,

which efficiently represented multipolygon data for graph classification tasks by leveraging visibility relationships between polygons. Specifically, PolygonGNN showed that augmenting vertex embeddings of individual polygons with the information of both spatial locations and visibility relationships to other polygon vertices is much more effective in capturing the geometric structure. All the above work assumes the presence of an underlying metric space or spatial position information which is absent in visibility graph reconstruction. We develop **VisDiff** to learn embeddings in this challenging combinatorial domain.

## 3 VISDIFF ARCHITECTURE

VisDiff consists of three main modules: Graph Encoding, SDF Representation Learning, and Vertex Prediction. The following sections focus on the details of each module.

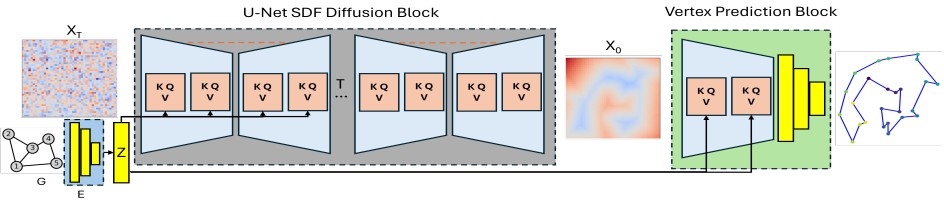

Figure 2: VisDiff architecture: There are three main blocks, namely U-Net SDF Diffusion, Vertex Prediction, and graph encoding. **G** represents a polygon structural graph. **E** represents the graph encoding module. **Z** represents the encoding of the graph. The noise added SDF after the forward diffusion process is represented by $\mathbf{X}_T$. **U-Net Diffusion Block**: $\mathbf{X}_T$ goes through **T** timesteps of reverse diffusion process to output the clean SDF represented by $\mathbf{X}_0$. **Vertex Prediction Block:** $\mathbf{X}_0$ is an input to the vertex prediction network, which generates the ordered vertex locations representing the polygon. The SDF and polygon generation are both conditioned on visibility through the cross-attention module. **K,Q**, and **V** are the key, query, and value terms of cross-attention. In our approach, **Q** is represented by the visibility embedding **Z** while **K** and **V** are represented by learned spatial CNN features. During **Training**: the model is supervised using both the ground truth SDF and polygon. During **Testing**: only the visibility graph **G** is provided as input.

**Graph Encoding:** The visibility graph is represented as a binary adjacency matrix. To condition other components of VisDiff on this input, we train a U-Net (Ronneberger et al., 2015) autoencoder with Binary Cross Entropy (BCE) Loss to reduce the dimensionality of the $25 \times 25$ (polygon with 25 vertex locations) input matrix to 512. We pretrain the autoencoder separately and freeze the encoder layer during encoding visibility graph $G$ in other modules.

**SDF Diffusion:** Diffusion models have shown the ability to efficiently learn the space of all images (Ramesh et al., 2022). Motivated by this success, we represent polygons with their signed distance functions which in turn can be represented as images (each pixel stores the distance to the nearest point on the polygon boundary). We can now learn the space of polygons as a diffusion process using a Denoising Diffusion Implicit Model (DDIM) (Song et al., 2020). DDIM primarily involves two steps: forward diffusion and the reverse diffusion processes.

*Forward Diffusion* process involves adding noise to the SDF representation in a scheduled manner. Let the SDF sample from the valid polygon distribution be denoted by $x_0 \in \mathbb{R}^{40 \times 40}$. Given the standard deviation of the noise level denoted by $\sigma_t > 0$ at timestep t of the diffusion step, the noise addition process is defined by $x_t = x_0 + \sigma_t \epsilon$ where $\epsilon \sim \mathcal{N}(0, I)$ is a sample from the Gaussian distribution. In this way, noise is continuously injected into the SDF, eventually transforming it into a pure Gaussian sample at the end of the forward noising process. VisDiff uses a linear log scheduler (Permenter & Yuan, 2023) to control the noise level throughout the forward noising process.

*Reverse Diffusion* involves recovering the original SDF from the final Gaussian sample generated during the forward diffusion process. In this step, we start with Gaussian noise and predict the noise added to the sample given the $\sigma_t$. The reverse diffusion is parameterized through a neural network that learns to predict the added noise given the input noise sample and $\sigma_t$.

Specifically, we train a U-Net (Ronneberger et al., 2015) encoder-decoder architecture to predict the noise added to the original SDF sample. Additionally, we condition the U-Net CNN blocks on encoded visibility using multiple Spatial Transformer Cross Attention (Ngo et al., 2023) blocks. The cross-attention blocks directly incorporate visibility information into the U-Net spatial features during the learning process. The key and value components of the cross-attention block are the spatial CNN features, while the query is the encoded visibility embedding. Figure 2 shows the architecture of the SDF Diffusion block. The model is trained using $L_{MSE}$ mean-squared error loss (MSE) between the predicted noise and the actual noise added to the sample. Given the visibility graph G, the trained model is then used to sample polygon SDF.

*Sampling* of the SDF is performed using a DDIM sampler. The sampling process draws a sample from a Gaussian distribution $\mathcal{N}(0, I)$ denoted by $x_t$ along with a schedule of decreasing noise levels proportional to the number of steps in the sampling process. Each diffusion step is given by Equation 1.

$$x_{t-1} = x_t + (\sigma_{t-1} - \sigma_t)\epsilon_\theta(x_t, \sigma_t, G) \tag{1}$$

where $\epsilon_\theta(x_t, \sigma_t, G)$ represents the noise predicted by the U-Net encoder-decoder architecture given the visibility graph G, the noise sample from the previous step $x_t$ and the standard deviation of the noise level $\sigma_t$. This process reconstructs the SDF of the polygon, ensuring it adheres to the visibility constraints defined by G.

**Vertex Prediction:** The generated SDF of the polygon is then used to determine the final vertex locations whose visibility relationship corresponds to the visibility graph G. The process of picking vertex locations over the zero level-set is challenging as the corners of the polygons are not well-defined in the SDF image. Furthermore, as the number of vertex locations increases, a small change in the placement of points on the SDF will significantly alter the visibility of the entire polygon.

We formulate the polygon vertex extraction as a separate estimation problem of determining vertex locations given the SDF and the visibility graph G. Specifically, we train a CNN encoder to encode the SDF into an embedding space. The embedding process is also conditioned on the visibility graph $G$ encoding using Spatial Transformer Cross Attention (Ronneberger et al., 2015) layers, which helps relate vertex generation to the visibility constraints. The keys and values for the spatial transformer are the spatial CNN features similar to the diffusion block, while the query is the encoded visibility embedding. The generated SDF embedding is then passed through multiple MLP layers to predict the ordered vertex locations of the polygon. See Figure 2.

We experimented with predicting vertex locations simultaneously with the SDF. Comparisons presented in Appendix (Section C, Table 11) show that training the vertex prediction model independently from the SDF generation model is significantly more accurate than joint training and prediction. Hence, we train the vertex prediction model separately with the ground truth SDF.

## 4 LOSS FUNCTIONS

The model is trained using the following loss function

$$Loss = \lambda_1 L_{MSE} + \lambda_2 L_{validity} + \lambda_3 L_{visibility} + \lambda_4 L_{SDF} \tag{2}$$

where $\lambda_i$ is a scaling factor and $\lambda_1 = \lambda_4 = 1.0$ while $\lambda_2 = \lambda_3 = 0.1$. $\lambda_2$ and $\lambda_3$ were chosen as 0.1 because the scale of $L_{Validity}$ and $L_{Visibility}$ is 10 times bigger than the other components. Each loss component has a unique role in learning the visibility property efficiently as described below.

$L_{MSE}$: The MSE loss penalizes deviation from ground truth vertex locations.

$$L_{MSE} = \|\hat{X} - X^*\|_2^2 \tag{3}$$

where $\hat{X}$ denotes the locations of the predicted vertices and $X^*$ denotes the ground truth vertex locations. $L_{MSE}$ loss is especially helpful for initial learning of the polygon structure.

$L_{visibility}$: The loss component $L_{visibility}$ measures how close the visibility graph $\hat{G}$ of the output polygon is to the input $G$ which can be computed using binary cross entropy.

$$L_{visibility} = L_{BCE}(\hat{G}, G) \tag{4}$$

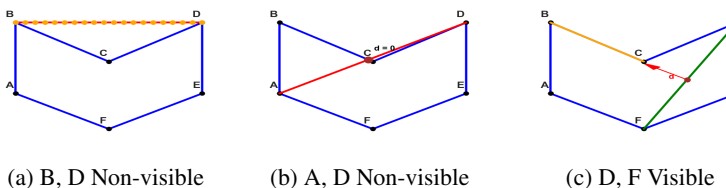

(a) B, D Non-visible      (b) A, D Non-visible      (c) D, F Visible

Figure 3: Visibility losses: To check whether two vertices $u$ and $v$ are visible to each other, we consider the intersection of the lines-segment $|uv|$ with the edges of the polygon and handle a degenerate case separately: In Figure 3a, the segment $BD$ does not intersect any polygon edge but it lies completely outside the polygon. The loss component $L_{out}$ addresses this case. For the remaining cases, the loss $L_{int}$ calculates $Int(X, Y) = 1/(1 + d(X, Y))$ where $X$ and $Y$ are two line segments and $d$ is the distance of the intersection point to the closest point on $X$. In Figure 3b, when $X = AD$ and $Y = BC$, the value of $d(AD, BC) = 0$ because the intersection point is on $AD$. However in Figure 3c, the value of $d(FD, BC) > 0$ as $FD$ and $BC$ are non-intersecting. $L_{int}$ calculates the $Int$ function with all polygon edges separately during the visibility calculation.

where $\hat{G}$ represents the predicted visibility graph and $L_{BCE}$ refers to binary cross entropy.

*However, since **VisDiff** outputs only vertex locations, the main challenge in computing this loss is computing the visibility graph in a differentiable manner.* We present a differentiable method to estimate $\hat{G}$. An edge is considered non-visible if it intersects any other polygon edge or is fully outside the polygon. We estimate $\hat{G}$ using two terms $L_{out}$ and $L_{int}$ to account for both conditions of non-visibility. See Figure 3. $L_{out}$ determines non-visibility due to being fully outside the polygon while $L_{int}$ determines non-visibility due to intersection. Specifically, $L_{out}$ samples dense points on the line and extracts the SDF values of points outside the polygon. $L_{int}$ calculates the distance to the intersection point between the visibility edge and each polygon edge. Equation 5 shows the resultant $\hat{G}$ for determining visibility for single edge $i$ given the $L_{int}$ and $L_{out}$.

$$\hat{G}_i = 1 - max(L_{intmax}, L_{outmax}) \tag{5}$$

where $L_{intmax}$ shows $max(L_{int})$ and $L_{outmax}$ shows $max(L_{out})$. We subtract one as non-visible edges are represented as 0 in the visibility matrix. $Max$ is a non-differentiable operation. We design a soft maximum to have a differentiable estimation of the maximum operation. Equation 6 shows a differentiable estimation of the maximum operation given two random numbers $A$ and $B$ where $softmax(A, B) = e^A/(e^A + e^B)$.

$$softmaximum(A, B) = softmax(A, B) \cdot (A + B) \tag{6}$$

The differentiable estimation of the maximum operation is used to determine each edge in $\hat{G}$.

$L_{validity}$: We introduce $L_{validity}$ to penalize polygon edge crossings. $L_{validity}$ uses the $Int$ function from $L_{visibility}$ to identify invalid configurations. Equation 7 shows the validity loss.

$$L_{\text{validity}} = \frac{1}{(m+1)^2} \sum_{i=0}^{m} \sum_{\substack{j=0 \\ j \neq i}}^{m} \text{Int}(P_i, P_j) \tag{7}$$

where $m$ denotes number of polygon $P$ edges and $i \neq j$ restricts the sum to edges that are neither adjacent nor the same. The function $Int()$ is illustrated in Figure 3.

$L_{SDF}$: The final loss components ensures that the vertices lie on the polygon boundary i.e. the zero level set.

$$L_{SDF} = \sum_{i=0}^{n} |S(V_i)| \tag{8}$$

where $V_i$ represents the $i$-th vertex location of polygon $P$, S represents its SDF value and n represents the number of vertices of polygon $P$.

Ablation studies in Appendix (Section C, Table 11) show that adding these additional losses helps the model improve on upholding the visibility graph G compared to training with only the $L_{MSE}$.

## 5 DATASET GENERATION

The problems of *Visibility Characterization* and *Visibility Reconstruction* require the dataset distribution to have a key characteristic of multiple polygons $P$ corresponding to the same visibility graph $G$. Additionally, the dataset should also represent a high diversity of visibility graphs. We address these characteristics by uniformly sampling polygons based on graph properties described below and also generate multiple augmentations of the same polygon.

The dataset generation process involves sampling 60,000 polygons with 25 vertex locations arranged in fixed anticlockwise ordering. The vertex locations are drawn from a uniform distribution within $[-1, 1]^2$. We use the 2-opt move (Auer & Held, 1996) algorithm to generate polygons from the drawn locations. We observed that the dataset generated from the 2-opt move algorithm exhibited non-uniformity with respect to the link diameter of the visibility graph. Link diameter quantifies the maximum number of edges on the shortest path between any two graph nodes. A higher diameter indicates greater concavity in the polygon. Hence, to have a balanced distribution, we resample the large dataset based on the link diameter of the visibility graph. The resampling process results in a subset of 18,500 polygons. In the appendix (Section D, Figure 8b) we present additional statistics showing that that our dataset is uniformly distributed in terms of link diameter.

We further augment each polygon to generate 20 samples by applying shear transformation and vertex perturbation while preserving the visibility graph $G$. The augmentations introduce the property of multiple polygons with the same visibility graph G . The augmentation and resampling are critical for learning the representative space of *Visibility Characterization* and *Visibility Reconstruction* problems. The final dataset consists of 370,000 polygons and their respective visibility graphs. The total dataset size including all polygons consists of 400,000, which will be made publicly available.

### 5.1 TEST SET GENERATION

We generate two datasets for evaluation: in-distribution and out-of-distribution. In-distribution samples are generated by setting aside 100 unique polygons per link diameter from the large dataset. These are not included in the training data.

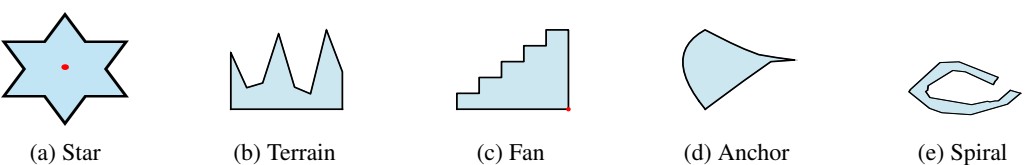

|  (a) Star  |  (b) Terrain  |  (c) Fan  |  (d) Anchor  |  (e) Spiral  |

Figure 4: Polygon types: a) **Star**: Single kernel point (red) from which all vertex locations are visible, b) **Terrain**: X-monotone polygons where orthogonal lines from the X axis intersect the polygon boundary at most twice, c) **Convex Fan**: Single convex vertex (red) which appears in every triangle of the polygon triangulation, d) **Anchor**: Polygons with two reflex links and a convex link connecting both of them, e) **Spiral**: Polygons with long link diameter.

The out-of-distribution samples are generated based on specific polygon types - star, spiral, anchor, convex fan, and terrain. Figure 4 details the properties of the polygon types. Spiral and anchor share similar characteristics to our dataset while terrain, convex fan and star differ significantly in terms of its density i.e., the total percentage of edges in the graph. In the appendix (Section D, Figure 9a) shows the difference in density of visibility graph distribution of terrain, convex fan, and star compared to the training set.

## 6 RESULTS

We evaluate VisDiff with baselines on the problem of *Visibility Reconstruction*. We also show the ability of VisDiff to give evidence for *Visibility Characterization* problem. We then provide preliminary results on *Visibility Recognition*. Lastly, we showcase the generalization of VisDiff to other graph structural properties like Triangulation.

## 6.1 EVALUATION METRICS

To evaluate our algorithm, we compute the visibility graphs of the output polygons and formulate the evaluation of the visibility graph as a classification problem. We report the accuracy, precision, recall, and F1-Score between the generated and the ground-truth visibility graphs. Specifically, each edge of the visibility graph is classified as either a visible or non-visible edge. Each visibility graph is evaluated individually, and the average over the dataset is reported as a collective quantitative metric. Since the ratio of visible and non-visible edges can be vastly different across polygons, we use the F-1 score to evaluate model performance.

## 6.2 QUALITATIVE AND QUANTITATIVE EVALUATION

We compare VisDiff against baselines, which generate vertex representation of a polygon from the visibility graph. In particular, we compare against various state-of-the-art encoders such as Transformer-Decoder [Seq] (Vaswani, 2017), Graph Neural Network [Gnn] (Veličković et al., 2017), DDIM [VD] (Song et al., 2020), Encoder - Decoder [E.D], VAE [VA] (Kingma & Welling, 2022) and a direct optimization approach based on Nelder-Mead [NM] (Gao & Han, 2012) optimization. Nelder-Mead optimizes the configuration of vertex locations by using the difference between the predicted and actual visibility graph as a loss which is backpropagated to the vertex locations. The code will be made publicly available for details on the implementation of all baselines.

### 6.2.1 *Visibility Reconstruction*

Table 1 shows the quantitative evaluation on the in-distribution dataset. VisDiff performs significantly better than architectures utilizing vertex representation on all metrics except for precision. Nelder-Mead optimization based on predicted and actual visibility graphs performs much better on precision, but it needs to be noted that it has the lowest recall as well. Specifically, Nelder-Mead optimization missed an average of 60% visible edges on all samples in the test dataset. Figure 5 also shows that Nelder-Mead optimization and others fail to generate valid polygons, ensuring both validity and visibility while VisDiff learns to generate polygons close to the ground truth visibility. We also provide additional quantitative results in Appendix (Section B).

|           | Acc ↑ | Prec ↑ | Rec ↑ | F1 ↑ | DAcc ↑ | DRec ↑ | DF1 ↑ | CDist ↓ |
|-----------|-------|--------|-------|------|--------|--------|-------|---------|
| (a) E.D   | 0.75  | 0.76   | 0.54  | 0.62 | 0.95   | 0.69   | 0.81  | 0.95    |
| (b) Seq   | 0.68  | 0.58   | 0.65  | 0.61 | 0.96   | 0.75   | 0.85  | 0.96    |
| (c) Gnn   | 0.73  | 0.90   | 0.43  | 0.57 | 0.95   | 0.70   | 0.82  | 1.03    |
| (d) VD    | 0.77  | 0.80   | 0.58  | 0.66 | 0.93   | 0.55   | 0.71  | 0.96    |
| (e) NM    | 0.70  | **0.93** | 0.34  | 0.49 | 0.98   | 0.88   | 0.94  | 1.10    |
| (f) Ours  | **0.85** | 0.83 | **0.77** | **0.80** | **0.99** | **0.95** | **0.97** | **0.91** |
| (g) VA    | 0.66  | 0.54   | 0.70  | 0.60 | 0.95   | 0.75   | 0.85  | 0.96    |

Table 1: Baseline comparison:(a) Encoder-Decoder, (b) Sequence Prediction, (c) GNN, (d) Vertex Diffusion, (e) Nelder-Mead Optimization, (f) VisDiff, (g) Variational Autoencoder, **Acc**: Accuracy, **Prec**: Precision, **Rec**: Recall, **DAcc**: triangulation accuracy, **DRec**: triangulation recall, **DF1**: triangulation F-1 Score, **CDist**: Chamfer distance between point sets in triangulation

We further evaluate VisDiff on its generalization to different polygon types. Table 2 shows its quantitative results on the out-of-distribution dataset. VisDiff generalizes well to polygons different from the training distribution. Specifically to the terrain, star, and convex-fan which have density of the visibility graph different from our distribution.

### 6.2.2 *Visibility Characterization*

We showcase the ability of VisDiff to present evidence for the *Visibility Characterization* problem. We generate multiple polygons given the same visibility graph $G$ by drawing different samples from Gaussian distribution for diffusion initialization. Figure 6 shows how VisDiff generates different polygons with perturbation and shear transformation but having similar visibility to the ground truth visibility graph $G$. The ability of sampling multiple polygons with the same visibility was also

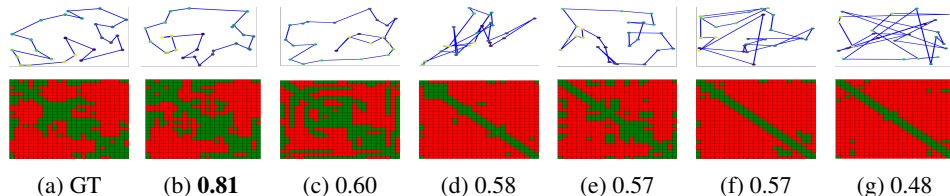

(a) GT  (b) **0.81**  (c) 0.60  (d) 0.58  (e) 0.57  (f) 0.57  (g) 0.48

Figure 5: Visibility reconstruction qualitative results: The top row shows the polygons generated by different methods. The first vertex is represented by deep purple and the last vertex by yellow (anticlockwise ordering). The second row shows corresponding visibility graphs of the polygons where **green** represents the visible edge and **red** represents the non-visible edge. The captions indicate the F1 Score of the visibility graph compared to the GT. The polygon results correspond to the following methods - a) Ground Truth, b) VisDiff c) Sequence Prediction d) GNN, e) Vertex diffusion, f) Encoder-Decoder, g) Optimization.

| Metrics | Accuracy ↑ | Precision ↑ | Recall ↑ | F1-Score ↑ |
|---------|-----------|-------------|----------|------------|
| Spiral | 0.875 | 0.842 | 0.808 | 0.823 |
| Terrain | 0.866 | 0.815 | 0.645 | 0.712 |
| Convex Fan | 0.769 | 0.775 | 0.772 | 0.771 |
| Anchor | 0.89 | 0.935 | 0.935 | 0.935 |
| Star | 0.772 | 0.751 | 0.797 | 0.77 |

Table 2: Specific polygon types: VisDiff shows generalization to star, terrain and anchor polygon types which are out of distribution samples to our dataset.

utilized in the above *Visibility Reconstruction* experiments. In particular, we sample 50 polygons given a single visibility graph $G$ and get the polygon best following the visibility graph $G$.

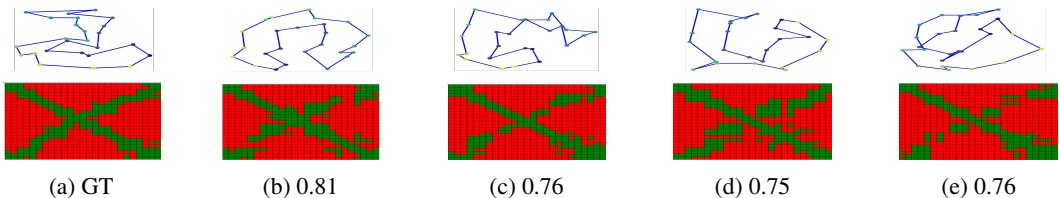

(a) GT  (b) 0.81  (c) 0.76  (d) 0.75  (e) 0.76

Figure 6: *Visibility Characterization*: The top row shows multiple polygons generated by VisDiff for the same visibility graph $G$. The first vertex is represented by deep purple and the last vertex by yellow (anticlockwise ordering). The second row shows the visibility graph corresponding to the polygons where **green** represents visible edge and **red** represents non-visible edge. The caption shows the F1-Score compared to the ground truth (GT) visibility graph.

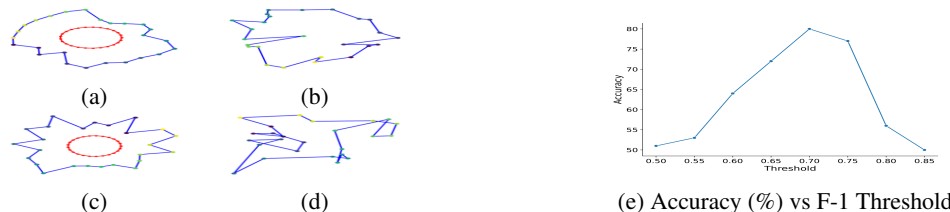

(a)  (b)

(c)  (d)  (e) Accuracy (%) vs F-1 Threshold

Figure 7: *Visibility Recognition*: a) Non-Valid Sample 1: Red represents hole, b) VisDiff Prediction Sample 1: VisDiff learns to put points in such a way to best maintain the visibility and the visibility graph is detected as a valid visibility graph, c) Non-Valid Sample 2: Red represents hole, d) VisDiff Prediction Sample 2: VisDiff failed to generate a valid polygon and therefore classified as a non-valid visibility graph, e) *Visibility Recognition* Quantitative Results: VisDiff classifies 80% of the samples correctly when the F-1 threshold is selected as 0.73.

### 6.2.3 *Visibility Recognition*

We present preliminary results on the *Visibility Recognition* problem. We generate a set of 50 valid and non-valid visibility graphs for the *Visibility Recognition* problem. We use polygons with holes as samples of non-valid visibility graphs. A polygon with a hole is a polygon with an outer boundary, but also has an inner boundary which makes it non simple. We determine the visibility graph in the same way as that of simple polygon. An edge through the hole is a non visible edge since the hole is considered outside the polygon.

We utilize the model's ability to sample multiple polygons and sample a set of polygons S for each visibility graph. If any of the polygons from S are valid and has a F1-Score over a certain threshold X, it is classified as a valid visibility graph. Figure 7e shows the performance of our model on *Visibility Recognition* problem using different thresholds on F-1 Score. Figure 15a to 15d shows qualitative results on the polygon generation for two non-valid visibility graphs. VisDiff is able to correctly classify 80% of the samples from the set of valid and non-valid visibility graphs when the F-1 threshold is selected to be close to mean performance on the *Visibility Reconstruction* problem. Classification performance of 80% shows that VisDiff is able to represent the underlying valid visibility graph space efficiently. Appendix E.4 shows more qualitative results on *Visibility Recognition*.

### 6.2.4 TRIANGULATION

In this section, we change the input from the complete visibility graph to the triangulation to show case the versatility of VisDiff. Note that a polygon may have many different triangulations. Each triangulation contains $n - 2$ triangles where $n$ is the number of vertices (De Berg, 2000). We use the Constrained Delauney Triangulation (Rognant et al., 1999) to triangulate the polygons in our dataset, ensuring a unique triangulation for a polygon (Dinas & Banon, 2014).

We evaluate the model on the classification metrics of the triangulation and the Chamfer distance (Borgefors, 1988). The classification metrics are calculated by comparing the existence of triangulation edges in the visibility graph of the generated polygon. In the case the model predicts a convex polygon given a triangulation of a non-convex polygon. It would have 100% triangulation accuracy which is misleading. Hence, the Chamfer distance between the points is also evaluated as the triangulation is unique to the spatial locations of the points. The Chamfer distance is calculated with polygons rotated to have the first edge aligned with the x-axis to account for rotation variations. Table 1 shows the quantitative results of VisDiff with baselines. VisDiff performs much better than all the models in maintaining the triangulation while also has the minimum Chamfer distance. We present additional qualitative results in Appendix (Section E.3, Figure 14)

## 7 CONCLUSION

In this paper, we studied the problems of Visibility Reconstruction, Characterization and Recognition for simple polygons. We presented **VisDiff** a diffusion-based approach which first predicts the Signed Distance Function (SDF) associated with a polygonal boundary conditioned on the input visibility graph $G$. The SDF is then used to generate vertex locations of a polygon $P$ whose visibility graph is $G$. Our method showed an improvement of 21% on F1-Score compared to baseline approaches on the *Visibility Reconstruction* problem. We then showed the capability of VisDiff to sample multiple polygons for a single visibility graph $G$ as a realization of *Visibility Characterization* problem. We also presented preliminary results of 80% accuracy on the *Visibility Recognition* problem. VisDiff has been shown to generalize to accept triangulations as input where it maintains 95% triangulation edges and achieves 4% improvement on Chamfer distance compared to baselines. We also proposed loss components for computing the visibility graph in a differentiable manner and demonstrated its effectiveness compared to guidance solely on L2 loss between vertex locations.

At a high-level, our results show that modern neural representations are capable of encoding the space of all polygons in such a way that the distances on the learned manifold are faithful to the combinatorial properties of polygons. In terms of future work, the presented VisDiff architecture represents the SDF as a grid, which creates a bottleneck in terms of computation time and space. In our future work, we will investigate encoding the SDF using more efficient representations such as (Park et al., 2019; Mitchell et al., 2020).

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

APPENDIX

# A  DEFINITIONS

In this section, we provide the definitions of the terms simple polygon and visibility graph.

| Terms | Definitions |
|---|---|
| Simple Polygon | Let $V = (v_1, \ldots, v_n)$ be an ordered set of $n$ points on the plane. The location of point $v_i$ is specified by its coordinates $(x_i, y_i)$. Let $e_i = (v_i, v_{i+1})$ be the set of line segments obtained by connecting consecutive points in $V$ in a cyclic manner. These line segments define a closed planar curve - the boundary of a polygon $P$. The points $v_i$ are the *vertices* of $P$ and the segments $e_i$ are its *sides*. Two consecutive edges of a polygon share an end-point at a vertex. In a simple polygon, these are the only intersections between the edges. The edges do not intersect each other. |
| Visibility Graph | A simple polygon $P$ has a well-defined interior and an exterior separated by its boundary $\delta P$. This separation allows us to define *visibility*: We will use the notation $x \in P$ to denote that $x$ lies either on the boundary or the interior of $P$. We say that two points $x, y \in P$ *see each other* if and only if $\forall z \in [x, y], z \in P$. In other words, the line segment $[xy]$ lies completely inside or on the boundary of $P$. The visibility graph of $P$, denoted $G(P)$ is a graph that is a vertex to vertex relation of $P$. There is an edge between two vertices $u$ and $v$ if and only if $u$ and $v$ are visible to each other in $P$. |

Table 3: Definitions

# B  QUANTITATIVE RESULTS

In this section, we present additional results on the evaluation of the SDF Diffusion model, a comparison of computational costs with the baseline, and the performance of the baseline models on the out-of-distribution test set.

## B.1  OUT-OF-DISTRIBUTION BASELINE PERFORMANCE

In this section, we provide the baseline performance on the out-of-distribution dataset for *Visibility Reconstruction* problem. Table 2 shows results of **VisDiff** while Table 4 - 8 shows the results of baselines on out-of-distribution dataset. A comparison of F-1 scores indicates that Visdiff performs significantly better than all the baselines on the out-of-distribution dataset.

## B.2  SDF DIFFUSION EVALUATION

We evaluate the SDF Diffusion model by measuring the L2 error between the ground truth and the predicted SDF on both in-distribution and out-distribution test datasets for the *Visibility Reconstruction* problem. Table 9 shows the performance of the SDF diffusion model. Our diffusion model predicts high-quality SDFs with low L2 error, indicating its effectiveness in capturing the underlying relationship between the polygon and visibility graphs.

| Metrics | Accuracy ↑ | Precision ↑ | Recall ↑ | F1-Score ↑ |
|---|---|---|---|---|
| Spiral | 0.712 | 0.602 | 0.602 | 0.6 |
| Terrain | 0.726 | 0.487 | 0.646 | 0.552 |
| Convex Fan | 0.569 | 0.584 | 0.573 | 0.574 |
| Anchor | 0.788 | 0.89 | 0.854 | 0.868 |
| Star | 0.575 | 0.558 | 0.588 | 0.568 |

Table 4: Specific polygon types: Sequence Prediction Performance

| Metrics | Accuracy ↑ | Precision ↑ | Recall ↑ | F1-Score ↑ |
|---|---|---|---|---|
| Spiral | 0.758 | 0.797 | 0.457 | 0.573 |
| Terrain | 0.8 | 0.653 | 0.578 | 0.602 |
| Convex Fan | 0.615 | 0.721 | 0.434 | 0.527 |
| Anchor | 0.698 | 0.929 | 0.694 | 0.791 |
| Star | 0.631 | 0.681 | 0.47 | 0.545 |

Table 5: Specific polygon types: Encoder-Decoder Performance

| Metrics | Accuracy ↑ | Precision ↑ | Recall ↑ | F1-Score ↑ |
|---|---|---|---|---|
| Spiral | 0.779 | 0.904 | 0.438 | 0.587 |
| Terrain | 0.85 | 0.841 | 0.541 | 0.654 |
| Convex Fan | 0.612 | 0.888 | 0.283 | 0.427 |
| Anchor | 0.452 | 0.865 | 0.419 | 0.553 |
| Star | 0.64 | 0.892 | 0.296 | 0.442 |

Table 6: Specific polygon types: GNN Performance

| Metrics | Accuracy ↑ | Precision ↑ | Recall ↑ | F1-Score ↑ |
|---|---|---|---|---|
| Spiral | 0.773 | 0.795 | 0.507 | 0.613 |
| Terrain | 0.843 | 0.79 | 0.553 | 0.646 |
| Convex Fan | 0.676 | 0.755 | 0.573 | 0.634 |
| Anchor | 0.756 | 0.887 | 0.817 | 0.849 |
| Star | 0.675 | 0.775 | 0.494 | 0.585 |

Table 7: Specific polygon types: Vertex Diffusion Performance

| Metrics | Accuracy ↑ | Precision ↑ | Recall ↑ | F1-Score ↑ |
|---|---|---|---|---|
| Spiral | 0.757 | 0.918 | 0.36 | 0.516 |
| Terrain | 0.851 | 0.909 | 0.478 | 0.625 |
| Convex Fan | 0.612 | 0.94 | 0.263 | 0.41 |
| Anchor | 0.298 | 0.98 | 0.176 | 0.299 |
| Star | 0.641 | 0.945 | 0.275 | 0.425 |

Table 8: Specific polygon types: Nelder-Mead Optimization Performance

| Test Dataset | L2 Error ↓ |
|---|---|
| In-Distribution | 0.071 |
| Out-Distribution: Spiral | 0.091 |
| Out-Distribution: Terrain | 0.091 |
| Out-Distribution: Convex Fan | 0.083 |
| Out-Distribution: Anchor | 0.158 |
| Out-Distribution: Star | 0.069 |

Table 9: SDF Evaluation: The table shows the L2 error between the predicted SDF from the diffusion model and the ground truth SDF

## B.3 COMPUTATIONAL COST COMPARISON

We compare the computational cost of our model with the baselines by evaluating the inference time per sample. The inference time of **VisDiff** is higher compared to baseline models. The increased inference time is because **VisDiff** performs the inference in two steps through the SDF while other baselines achieve it in a single step. Furthermore, GNN, Sequence Prediction, and Encoder-Decoder generate only one sample per visibility graph while **VisDiff** and vertex diffusion generate 50 samples per visibility graph.

| Baselines | Computational Time (seconds) $\downarrow$ |
|---|---|
| Encoder-Decoder | 0.001 |
| Sequence Prediction | 0.075 |
| GNN | 0.005 |
| Optimization | 74.210 |
| Vertex Diffusion | 0.094 |
| **VisDiff** | 1.02 |
| Variational Autoencoder | 0.003 |

Table 10: Computational Cost Comparison: Each inference time corresponds to the time in seconds taken for each model to generate vertex locations for a single visibility graph

## C ABLATION STUDIES

The two main directions of ablation studies performed for VisDiff are in loss functions and architecture choices. Table 11 shows the results achieved for different architecture choices. It shows that the best results are achieved by estimating the SDF and vertex locations separately. We also evaluate the change in performance with an addition of each component of loss. Table 12 shows that with the addition of all the loss components helps gain 10% F1-Score than using just the vertex locations error.

| | Accuracy $\uparrow$ | Precision $\uparrow$ | Recall $\uparrow$ | F1 $\uparrow$ |
|---|---|---|---|---|
| b) Joint | 0.78 | 0.80 | 0.60 | 0.68 |
| d) Separate | **0.85** | **0.83** | **0.77** | **0.80** |

Table 11: Ablation Studies: (a) Joint estimation of SDF with vertex locations, (b) Separate estimation SDF with vertex locations (VisDiff). The results are on *Visibility Reconstruction* problem

| | Accuracy $\uparrow$ | Precision $\uparrow$ | Recall $\uparrow$ | F1 $\uparrow$ |
|---|---|---|---|---|
| a) $L_{MSE}$ | 0.83 | 0.74 | 0.72 | 0.73 |
| b) $L_{MSE} + L_{Vis}$ | 0.84 | 0.83 | 0.70 | 0.76 |
| c) $L_{MSE} + L_{Vis} + L_{Val}$ | 0.85 | 0.83 | 0.73 | 0.78 |
| d) $L_{MSE} + L_{Vis} + L_{Val} + L_{SDF}$ | **0.85** | **0.83** | **0.77** | **0.80** |

Table 12: Ablation Studies Loss Components: (a) MSE Loss, (b) Adding Visibility Loss component, (c) Adding Visibility and Validity Loss component, (d) Adding Visibility, Validity and SDF Loss component. The results are on *Visibility Reconstruction* problem

## D DATASET STATISTICS

In this section, we present statistics about our dataset. Figure 8 shows the distribution of the train and in-distribution test set statistics. It shows that our dataset is uniform in diameter of the visibility

graph. Figure 9 compares the training dataset with the out-of-distribution testing dataset. It shows that star, convex-fan, and terrain classes have densities different from our train distribution, where density refers to the percentage of edges in the visibility graph.

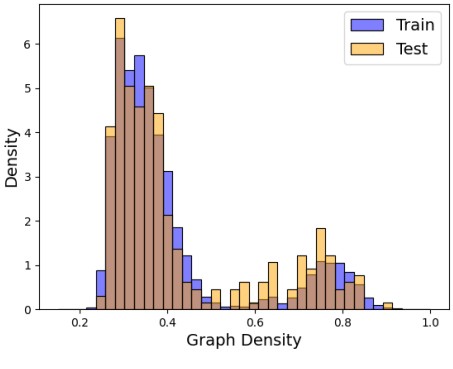 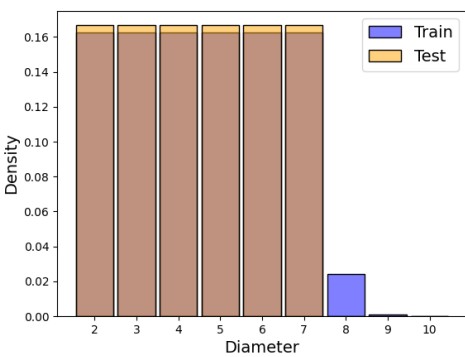

(a) Density Comparison Train vs Test      (b) Diameter Comparison Train vs Test

Figure 8: Train vs in-distribution test set analysis: 8a) The density is inversely proportional to the diameter. Uniform sampling of diameter results in bimodal density. 8b) Training and testing sets are uniform in terms of the link diameter of the visibility graph.

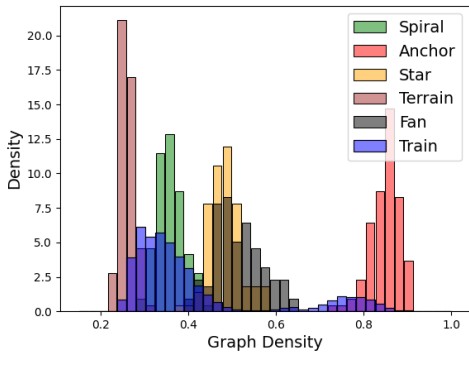 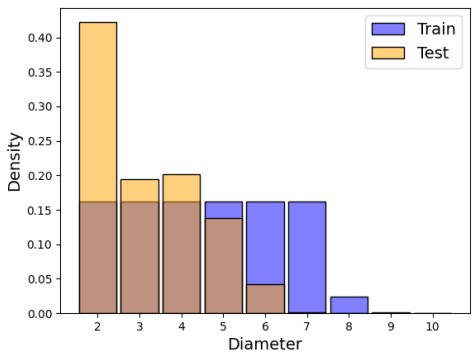

(a) Density Comparison Train vs Test      (b) Diameter Comparison Train vs Test

Figure 9: Out-of-distribution test set analysis: Figure 9a shows the density of the anchor and spiral are close to the mean of the bimodal training distribution, making it similar to our training set. The density of the star, convex fan, and terrain differ significantly from the training distribution.

# E  QUALITATIVE RESULTS

In this section, we provide additional qualitative results on *Visibility Reconstruction*, *Visibility Characterization*, *Visibility Recognition*, and the Triangulation problem (Section 6.2.4).

## E.1  *Visibility Reconstruction*

We provide additional qualitative results for the *Visibility Reconstruction* problem. Figures 10 and 11 show the comparison between polygons generated by VisDiff to baselines. The F1-Score shows that VisDiff generates polygons much closer to the visibility graph of the ground truth polygon.

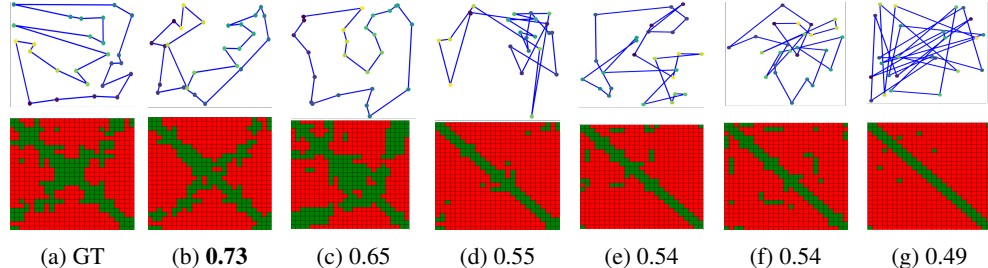

(a) GT  (b) **0.73**  (c) 0.65  (d) 0.55  (e) 0.54  (f) 0.54  (g) 0.49

Figure 10: Visibility reconstruction qualitative results: The top row shows the polygons generated by different methods. The first vertex is represented by deep purple and the last vertex by yellow (anticlockwise ordering). The second row shows corresponding visibility graphs of the polygons where **green** represents the visible edge and **red** represents the non-visible edge. The polygon results correspond to the following methods - a) Ground Truth, b) VisDiff  c) Sequence Prediction d) GNN, e) Vertex diffusion, f) Encoder-Decoder, g) Optimization. The captions indicate F1-Score

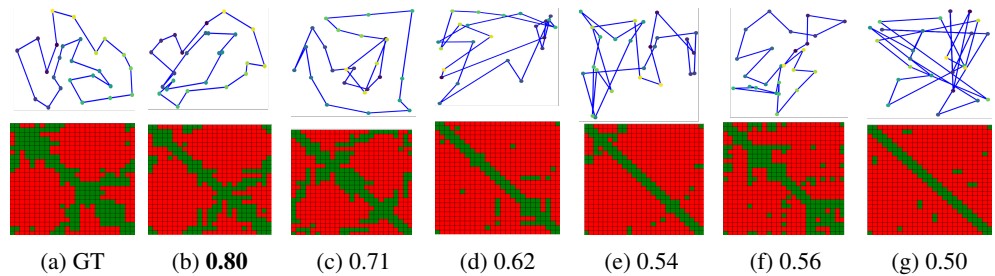

(a) GT  (b) **0.80**  (c) 0.71  (d) 0.62  (e) 0.54  (f) 0.56  (g) 0.50

Figure 11: Visibility reconstruction qualitative results: The top row shows the polygons generated by different methods. The first vertex is represented by deep purple and the last vertex by yellow (anticlockwise ordering). The second row shows corresponding visibility graphs of the polygons where **green** represents the visible edge and **red** represents the non-visible edge. The polygon results correspond to the following methods - a) Ground Truth, b) VisDiff  c) Sequence Prediction d) GNN, e) Vertex diffusion, f) Encoder-Decoder, g) Optimization. The captions indicate F1-Score

### E.2 *Visibility Characterization*

We provide further qualitative results on the problem of *Visibility Characterization* where we seek to generate the set of all polygons associated with the same visibility graph. Figures 12 and Figure 13 show the ability of VisDiff to sample multiple polygons given same visibility graph.

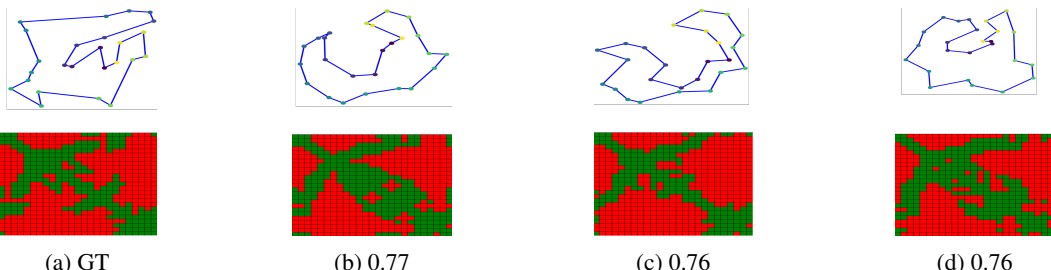

(a) GT  (b) 0.77  (c) 0.76  (d) 0.76

Figure 12: *Visibility Characterization*: The top row shows multiple polygons generated by VisDiff for the same visibility graph $G$. The first vertex starts at deep purple and the last vertex ends at yellow (anticlockwise ordering). The second row shows the visibility graph corresponding to the polygons where **green** represents visible edge and **red** represents non-visible edge. Subfigure captions indicate the F1-Score

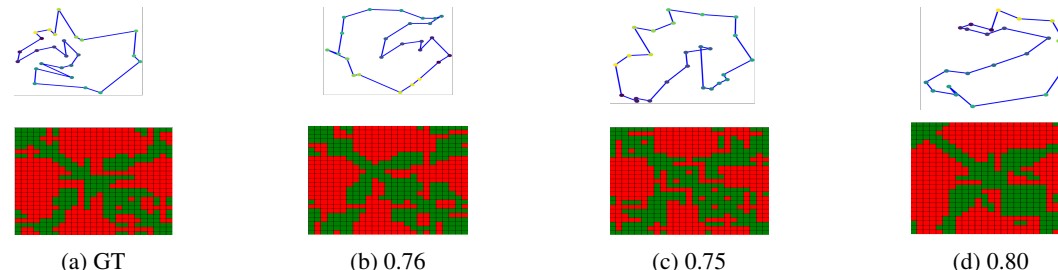

(a) GT        (b) 0.76        (c) 0.75        (d) 0.80

Figure 13: *Visibility Characterization*: The top row shows multiple polygons generated by VisDiff for the same visibility graph $G$. The first vertex is represented by deep purple and the last vertex by yellow (anticlockwise ordering). The second row shows the visibility graph corresponding to the polygons where **green** represents visible edge and **red** represents non-visible edge. Subfigure captions indicate the F1-Score

### E.3 TRIANGULATION

We provide qualitative results for the problem of generating polygons from the triangulation. Figure 14 shows the performance of VisDiff compared to other baselines. VisDiff maintains 98% of the triangulation edges.

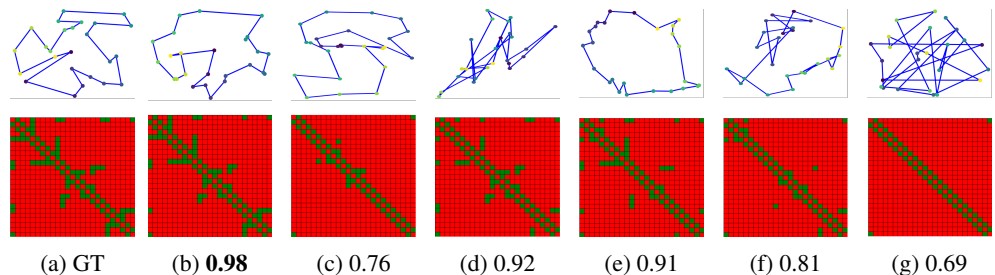

(a) GT    (b) **0.98**    (c) 0.76    (d) 0.92    (e) 0.91    (f) 0.81    (g) 0.69

Figure 14: Triangulation Qualitative Results: Top row shows the polygons generated by different methods. The first vertex is represented by deep purple and the last vertex by yellow (anticlockwise ordering). The second row shows corresponding triangulation graphs of the polygons where **green** represents the triangulation edge and **red** represents the absence of the triangulation edge. The captions indicate the F1 Score of the triangulation graph compared to the GT. The polygon results correspond to the following methods - a) Ground Truth, b) VisDiff c) Sequence Prediction d) GNN, e) Vertex diffusion, f) Encoder-Decoder, g) Optimization

### E.4 *Visibility Recognition*

We provide additional qualitative results to showcase failure and successful instances of VisDiff on *Visibility Recognition* problem. Figure 15 shows the output of VisDiff when the input is not a valid polygon (We generate visibility graphs of polygons with holes as invalid input samples). It shows that VisDiff can be used to identify non-valid visibility graphs in most of the scenarios by turning it into a classifier based on the validity of the output.

## F VARIATIONAL AUTOENCODER

In this section, we present the results for the variational autoencoder baseline. Specifically, we include out-of-distribution dataset results for visibility reconstruction, along with qualitative results for both the visibility reconstruction and triangulation problems. Table 13 reports the out-of-distribution test set performance of the variational autoencoder. Figure 16 shows the qualitative comparison of the variational autoencoder and VisDiff on the *Visibility Reconstruction* task. Figure 17 illustrates a qualitative comparison for the Triangulation problem.

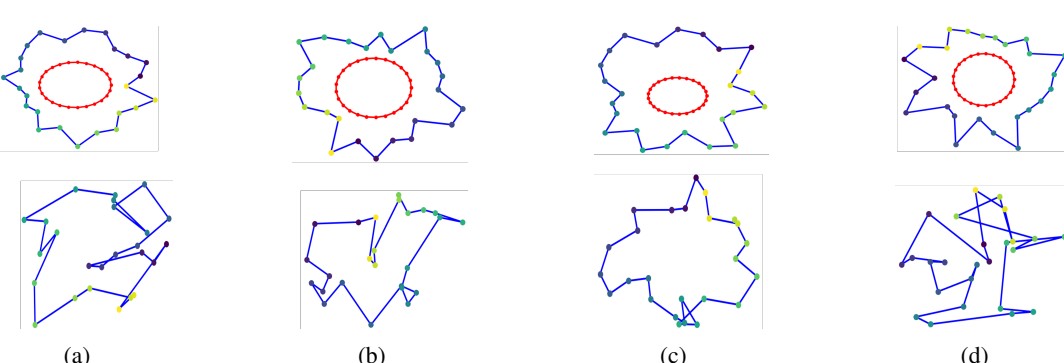

|     |     |     |     |
|-----|-----|-----|-----|
| (a) | (b) | (c) | (d) |

Figure 15: *Visibility Recognition*: The top row signifies the ground truth non-valid polygon with the hole (red) while the bottom row is the polygons drawn by VisDiff. The first vertex is represented by deep purple and the last vertex by yellow (anticlockwise ordering). a) Non-Valid Sample 1: VisDiff predicts it as a non-valid polygon as it is not able to generate any valid polygon, b) Non-Valid Sample 2: VisDiff generates valid polygon where it learns to put points in a $V$ shape to account for a hole. It misclassified a non-valid visibility graph as a valid visibility graph. c) Non-Valid Sample 3: VisDiff predicts it as a non-valid polygon as it is not able to generate any valid polygon, d) Non-Valid Sample 4: VisDiff predicts it as a non-valid polygon as it is not able to generate any valid polygon

| Metrics | Accuracy ↑ | Precision ↑ | Recall ↑ | F1-Score ↑ |
|---------|-----------|-------------|----------|------------|
| Spiral | 0.702 | 0.602 | 0.592 | 0.6 |
| Terrain | 0.706 | 0.467 | 0.626 | 0.534 |
| Convex Fan | 0.549 | 0.564 | 0.543 | 0.553 |
| Anchor | 0.768 | 0.87 | 0.834 | 0.851 |
| Star | 0.535 | 0.538 | 0.568 | 0.552 |

Table 13: Specific polygon types: Variational Autoencoder Performance

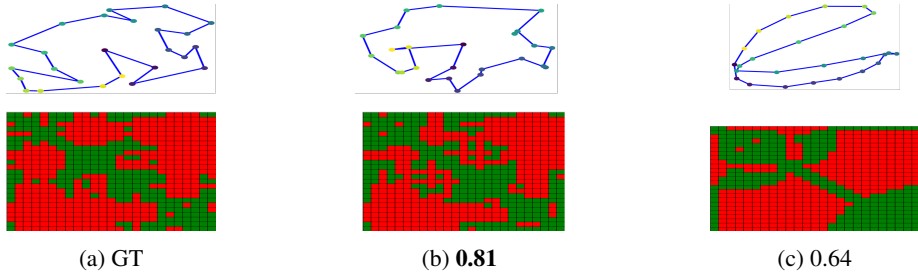

|     |     |     |
|-----|-----|-----|
| (a) GT | (b) **0.81** | (c) 0.64 |

Figure 16: Visibility reconstruction qualitative results: The top row shows the polygons generated by different methods. The first vertex is represented by deep purple and the last vertex by yellow (anticlockwise ordering). The second row shows corresponding visibility graphs of the polygons where **green** represents the visible edge and **red** represents the non-visible edge. The captions indicate the F1 Score of the visibility graph compared to the GT. The polygon results correspond to the following methods - a) Ground Truth, b) VisDiff c) Variational Autoencoder

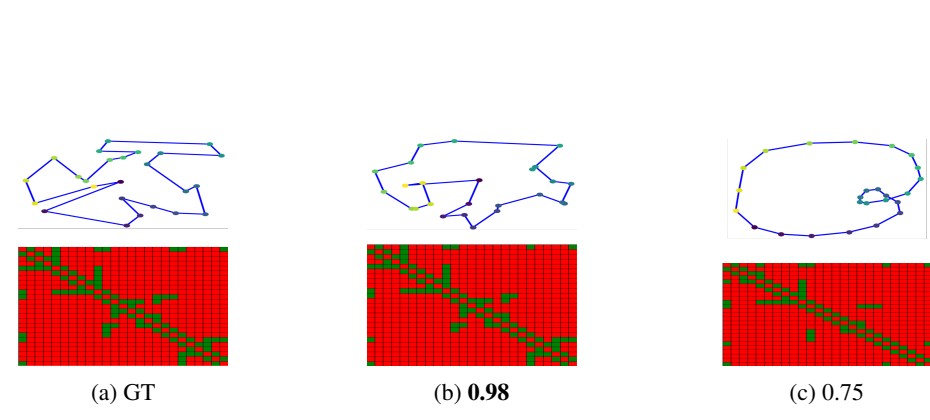

(a) GT        (b) **0.98**        (c) 0.75

Figure 17: Triangulation Qualitative Results: Top row shows the polygons generated by different methods. The first vertex is represented by deep purple and the last vertex by yellow (anticlockwise ordering). The second row shows corresponding triangulation graphs of the polygons where **green** represents the triangulation edge and **red** represents the absence of the triangulation edge. The captions indicate the F1 Score of the triangulation graph compared to the GT. The polygon results correspond to the following methods - a) Ground Truth, b) VisDiff c) Variational Autoencoder

