# OpenReview forum: "VisDiff: SDF-Guided Polygon Generation for Visibility Reconstruction and Recognition"
_ICLR.cc/2025/Conference — Submitted to ICLR 2025_

### Official Review · Reviewer_k2Tx · 2024-10-17

**Soundness:** 2
**Presentation:** 2
**Contribution:** 2
**Rating:** 6
**Confidence:** 2

**Summary:**

This paper learns latent representations for combinatorial structures focusing on polygons. It introduces "VisDiff," a diffusion-based model for reconstructing polygons from their visibility graphs. This process involves estimating the signed distance function (SDF) of a polygon from its visibility graph and extracting vertex locations in an ordered sequence that maintains the visibility relationships indicated by the graph's edges. This approach is enhanced by the integration of loss functions that enable the computation of visibility in a differentiable manner, alongside the creation of a specialized dataset for training.

**Strengths:**

1. The research problem addressed in this paper is of significant importance
2. The organization of the content is clear and logical, facilitating an easy understanding of the complex concepts presented.

**Weaknesses:**

1.1 The presentation of the paper could be improved for clarity and flow. In particular, the concept of "dual" introduced in line 51 is unclear. It would be helpful if the authors could explain whether this "dual" has a one-to-one correspondence with polygons, how it is derived, and its relationship to the polygons discussed.

1.2. The definition of the graph G also lacks clarity. Figure 1 seems to depict three different types of graphs labeled as "G", which is confusing. Could the authors specify what each graph represents? And in line 76 authors mentioned “G can be visibility graph or triangulation dual of the polygon”.  Are the visibility graph and the triangulation dual equivalent? how they are related?

2. The motivation behind the paper is currently not compelling. The introduction lists problems that appear purely mathematical with unclear real-world applications. The reasoning behind the inclusion of 'dual' and the visibility graph remains unclear.
3. Some relevant works are not discussed. For instance, [1] proposes modeling polygons using visibility graphs and a graph neural network for learning representations.
4. The experimental section raises concerns about reproducibility and the selection methodology. In line 375, the authors mention that the visibility graph is not unique for a given polygon. It is crucial to explain how the experimental process was standardized to ensure reproducibility. Moreover, considering the paper's objective to reproduce polygons, why not evaluate the effectiveness directly from the polygons instead of using a proxy? The high F1 score reported in Figure 6 does not seem to correspond well with the visual similarity of the generated polygons to the ground truth.

5.1 Technical details need clarification. For example, in line 188, the reasoning behind the choice of a 25*25, 512 configuration should be explained. What is the size of x and the noise in line 198?

5.2. The introduction of a "differentiable" max operation is intriguing. Since max is commonly used in machine learning and supported by frameworks like PyTorch, what specific advantages does your design offer?

[1] PolygonGNN: Representation Learning for Polygonal Geometries with Heterogeneous Visibility Graph. KDD '24: Proceedings of the 30th ACM SIGKDD Conference on Knowledge Discovery and Data Mining
Pages 4012 - 4022 https://doi.org/10.1145/3637528.3671738

**Questions:**

Refer to the weakness section for questions.

---

> ### Author Response · Authors · 2024-11-19
> **Reply to Reviewer k2Tx (Part 1/2)**
>
> Thank you for your overall positive evaluation and helpful feedback. The reviewer’s main concerns were about the **dual derivation and uniqueness, definition of G in Figure 1, Practical Applications, visibility graph uniqueness for reproducible evaluation, additional reference for related work, Evaluation metric, visual similarity in Figure 6, and differentiable max operation**. We address each of the reviewer’s concerns as follows-
>
> * Concern regarding the “dual” term **(W1.1)**
>   * The reviewer expressed confusion about the concept of the “dual” on how it is obtained and its uniqueness. To address the reviewer’s comment,  we obtained it using constrained Delaunay triangulation which is unique to the input polygon. Please refer to **Section 6.2.4** in the updated manuscript where we mention the following additional detail.
>
> * Additional reference for related work **(W3)**
>   * The reviewer mentions a reference which they think is relevant to our work. We have added our additional contributions to them in the related work section. We would like to thank the reviewer for suggesting the following reference.
>
> * Confusion regarding “G can be visibility graph or triangulation dual of the polygon” **(W1.2 and W2)**
>   * The reviewer expressed confusion about the reason for the inclusion of the visibility graph and triangulation dual in our work. We reiterate that our motivation was to study polygon reconstruction and recognition as a representative problem for representation learning of combinatorial structures that lack well-behaved local neighborhoods or distance functions. We say “G can be a visibility graph or triangulation dual of the polygon” to give some examples of such combinatorial structures. Furthermore, they are also related to each other as the triangulation dual edges are a subset of the edges of the visibility graph.
>
> * Figure 1 Clarity **(W.1.2)**
>   * The reviewer expressed confusion about the definition of G in Figure 1. All G in Figure 1 represent the same dual graph. We have modified Figure 1 to overlay graph G on the hand to make it more clear.
>
> * Practical Applications **(W2)**
>   * The reviewer questions the practical applicability of the approach. We acknowledge the fact that the approach does not have a ready practical application. We aimed to look at it from the perspective of representation learning for combinatorial structures lacking well-behaved local neighborhoods or distance functions. Although we briefly discussed the potential application of generating meshes from skeletal structures, we acknowledge that a detailed exploration of this application remains limited.
>
> * Uniqueness of Visibility graph for reproducibility of evaluation **(W4)**
>   * The reviewer is concerned about the reproducibility of the evaluation as they think the visibility graph is not unique to a polygon. We would like to respectfully correct the reviewer that visibility graphs are unique for a given polygon. We would also like to clarify that Line 375 does not comment on the non-uniqueness of the visibility graph for a single polygon. Visibility graphs define vertex-to-vertex relations, which remain constant unless the polygon is perturbed. All evaluations were performed on the final generated polygons by extracting their visibility graphs and comparing them with the given visibility graphs. Hence, the evaluations done in our work are reproducible. We have also added definitions in **Appendix A** of the updated manuscript for a better explanation of the visibility
>
> * Evaluation metric **(W4)**
>   * The reviewer questions why the polygon vertex locations comparison is not used as an evaluation metric. Multiple polygons can share the same visibility graph (characterization problem). Hence, directly comparing vertex locations is not an effective evaluation metric.
> *  Visual similarity between the ground truth polygon and the generated polygon in Figure 6 **(W4)**
>     * The reviewer is concerned that the high F1 score in Figure 6 does not align with the visual similarity between generated polygons and the ground truth. The difference in visual similarity is because multiple polygons (by perturbing the ground truth while keeping the visibility constant) can share the same visibility graph which is also our characterization problem. Hence, the polygon need not be necessarily visually similar to the ground truth polygon to have a high F1 score between the visibility graph.

---

> ### Author Response · Authors · 2024-11-19
> **Reply to Reviewer k2Tx (Part 2/2)**
>
> * 'Differentiable' max operation (W5.2)
>     * The reviewer is concerned about the use of the differentiable max operation as standard Pytorch max operation is differentiable. By 'non-differentiable', we mean the lack of smooth gradient flow from all components which the Pytorch max function does not offer. To address this, we introduced a 'differentiable' max operation that allows gradients to flow from all components, weighted by their importance, ensuring a more informative loss function. We hope this clarifies the concept of 'differentiable' max operation.
>
> We would like to finally thank the reviewer for their time and efforts in reviewing our work and giving constructive feedback. Please let us know if there are any further questions or clarifications, we would be happy to address them.

---

> > ### Comment · Reviewer_k2Tx · 2024-12-03
> >
> > Thank you for your reply, I have increased the score accordingly.

---

### Official Review · Reviewer_8j9a · 2024-11-04

**Soundness:** 3
**Presentation:** 4
**Contribution:** 3
**Rating:** 8
**Confidence:** 2

**Summary:**

This paper considers the following problem: Given a combinatorial structure such as a graph representing either the visibility graph or a triangulation, construct a polygon whose visibility graph/triangulation is this input graph G. Note that the coordinates of vertices are not provided in the input. This is the problem of visibility reconstruction. The problem is known to be incredibly hard and lies somewhere between NP and PSPACE.

The key contribution of this paper is to introduce a diffusion based method to reconstruct a polygon from its visibility graph. There are two main contribution of this paper. First, instead of generating the polygon directly, they first generate the signed distance function using which they then extract the vertex locations. The second contribution is in the training process where the authors design a novel loss function as well as create a curated data set. Combined together, they produce (both quantitatively and qualitatively) better results for this problem.

**Strengths:**

* Applying ML methods to significantly improve state-of-the-art on a hard problem such as polygon generation is important.
* The paper is extremely well-written and I was able to appreciate the various intricate issues that arise in solving this problem

**Weaknesses:**

I did not find any major weakness.

My confidence in evaluation is lower only because I’ve not explicitly worked in this space and may not be aware of all the existing work.

**Questions:**

NA

---

> ### Author Response · Authors · 2024-11-19
> **Reply to Reviewer 8j9a**
>
> Thank you for your overall positive evaluation. We sincerely appreciate your recognition of the novelty of our work.

---

### Official Review · Reviewer_hNLw · 2024-11-06

**Soundness:** 1
**Presentation:** 2
**Contribution:** 2
**Rating:** 3
**Confidence:** 3

**Summary:**

This paper introduces a diffusion-based approach for polygon reconstruction, characterization and recognition, which is guided by signed distance function (SDF). The main contributions lie in a novel differentiable loss function and data set with variable link diameter.

**Strengths:**

(S1) I really like the Introduction paragraph on lines 075-100 that very nicely describes the main goal and problems studied in the paper.

(S2) I also appreciate that you generated the in- and out-of-distribution polygons that bear some significant differences, and that the test data your experiments considers both cases.

**Weaknesses:**

(W1) Strength of contributions: At some moments in the manuscript, it seems that the authors imply that (at least some of) the novelty of the paper is related to diffusion and SDF. For instance:

- Abstract: "We introduce VisDiff, a novel diffusion-based approach ..."
- Abstract: "Our main insight is that going through the SDF significantly improves learning..."
- Introduction: "Our main insight is that going through the SDF as an intermediate representation yields superior results over using established methods to predict the vertex locations directly"
- Section 3 (when motivating the use of diffusion models): "Diffusion model have shown the ability to efficiently learn the space of all images" (without a reference), which seems to imply that their use in the context of polygon reconstruction is novel.

However, when looking at the list of references included in the paper, I see that the diffusion models have already been used in this context, and that seemingly more complicated settings such as pseudo-polygons, polygons with holes have been studied (whereas this paper focuses on the very basic setting of simply-connected polygons with boundary that does not self intersect). Moreover, a quick Google Scholar search makes it clear that both diffusion and SDF have been used in this area of research.

On the other hand, at different moments in the paper, the authors state that the main contributions lie in the novel loss function and a carefully designed data set of polygons. If this is the case, then the Related work section needs to focus or at least discuss these aspects as well, and the experiments need to evaluate their added value better, see see (Q4), (W2) and the related (Q7) below. What I then find confusing is that these two aspects are never mentioned in the Conclusion either, whereas the diffusion and SDF are discussed.

Finally, in the Conclusions, the authors write: "Overall, our results provide evidence that recent architectures can learn representations of non-trivial combinatorial structures as polygons", but I assume that this has been done before too? Also, such a formulation implies that this is your main contribution?

Overall, the above shows that some parts of the paper need to be rewritten to make it much clearer where the main contributions lie, what has been done before and in what way.

(W2) Quality of experiments: The paper has no theoretical results, and the experiments are not extensive enough. If the main contribution is the novel loss function, a more extensive ablation study is needed, that evaluates the added value of each of the components (whereas at the moment, you only consider the visibility loss), see (Q7). If the other main contribution is the data set, I wonder to what extent is it interesting to consider only polygons with 25 vertices? Would a more diverse data set with different number of vertices not make it possible to better differentiate the power of different models? Also, I find it rather striking that the other baseline state-of-the-art methods perform so poorly (e.g., in Figure 5), which makes me wonder if the data set you curate is not in some way specifically tailored to the proposed method. Would it not be fair to compare the performance of your and the baseline methods also on other data sets that are studied in those other papers? A quick look makes me think that these other baselines are developed to tackle completely different tasks, it is not then fair to claim that these are state-of-the-art methods for polygon reconstruction (that you outperform)? Why not compare with some other polgyon reconstruction approaches, and also using other existing data?

**Questions:**

(Q1) Is there a reason why characterization is not mentioned in the title too? You make such a nice distinction between reconstruction, characterization and recognition (see S1 above), but you only mention two of them in the title, whereas your experiments consider all three problems? Similar formulations come back at other moments in the paper too, e.g., first sentence in the Conclusion.

(Q2) How did you get the graph in Figure 1 right with Isomap, i.e., how did you decide which vertices are connected with an edge? Also, what do the different colors of the vertices represent? Related to (Q1), could this figure also include the visibility graph of P? Note also that you use notation G for three different things in the three plots of this figure, this should be made more precise? Instead of "Object", should you not use "P"?

(Q3) Do you truly address the problem of polygon characterization?  On line 111, you seem to imply that this is indeed the case, "we can sample *the* set of polygons which have a given visibility graph". However, in Section 6.4 on visibility characterization, you write "We generate multiple polygons given the same visibility graph G by drawing different samples from Gaussian distribution for diffusion initialization." Are you in this way generating *some/many*, or *all* polygons with the desired property? This needs to be made very precise.

(Q4) Is the Related work not missing the optimization (integer programming and heuristic) approaches? More importantly, and related to (W1) above, if the main contributions of the paper lie in the novel loss function and data set, please include these in your discussion. What types of loss functions have been used in the literature, have any comparisons been done so far? What other data sets are used, are there benchmarks, what are some their properties?

(Q5) Can you motivate your choices for the VisDiff architecture, e.g, why do you opt for U-Net or Spatial Transformer Cross Attention? Can Figure 2 explicitly label the three main modules, i.e., also the graph encoding / U-Net and SDF (and possibly also DDIM)? Should the caption, or at least the main text, not explain what X_T, X_9, K, Q, V are? As it is, the figure is not helping me understand the architecture.

(Q6) Why is your approach not compared against other methods also for the out-of-distribution data (Table 2)?

(Q7) What kind of loss function is typically used in the literature, is it only L_MSE, see related (Q4) above? This will help to clarify wherein lies the novelty of your approach, i.e., if you are the first to introduce L_validity, L_visibility and L_SDF, or is it in the differentiability i.e. approximating max with softmaximum. In the former case, the ablation study needs to assess the added value of each of the components. Moreover, you consider the same weight for each of the four components in the loss function, but if this is the main contribution of the paper, would it not be particularly interesting to consider different weight schemes?


Minor comments:
- For better readability, the order in which different notions are listed throughout the paper can be improved. For example, a more logical order in the first sentence of Figure 2 caption would be graph encoder (U-Net), denoising U-Net and vertex prediction. As another example, consider the order of models in Table 1 and Figure 5 (for better clarity, it would also be better if the model acronyms were included in the figure itself). Also, the order of plots in Figure 9 is opposite from Figure 8.
- line 188: At this moment, the reader is not yet familiar with the data set, i.e., with the fact that polygons have 25 vertices, so this should at least briefly be mentioned in brackets.
- line 228: image.Furthemore -> image. Furthermore
- There are some issues with numbering or formatting of the section titles. The section and subsection titles are mostly capitalized, but then this is not the case for Section 6.3, 6.4, 6.5 - are these maybe supposed to be subsections 6.2.1, 6.2.2 and 6.2.3?
- Explain in Figure 5 caption why the F1 scores do not match the F1 scores from Table 1.
- line 496: X. It -> X, it
- line 539: more efficient representations such as (Park et al, Mitchell et al). Can you describe in a few words what these representations are, or what the main idea/ingredient is?
- Check formatting in the references, e.g., capitalization of journals, Polydiff, 3d, Sd, 3dgen, Deepsdf, delaunay ...

---

> ### Author Response · Authors · 2024-11-19
> **Reply to Reviewer hNLw (Part 1/2)**
>
> We thank the reviewer for helpful suggestions and comments. We believe we were able to address most of them and hope that you will reconsider your low score. The reviewer’s main concerns were about the **Strength of the contribution, Dataset, Baseline, Loss function, missing important contribution in the conclusion and title, isomap polygon connectivity, Figure 2 clarity, Figure 1 clarity, polygon characterization and baseline comparison for out-of-distribution**. We address each of the reviewer’s concerns as follows-
>
> * Baseline comparison for out-of-distribution data **(Q6)**, loss function ablation studies **(Q7, W2)**, missing important contributions in the conclusion and title **(Q1)**, Figure 2 clarity **(Q5)**, Figure 1 clarity **(Q2)**.
>   * The reviewer mentions the addition of loss contribution, inclusion of characterization in the title, and recognition in the conclusion. The additions have been made to the revised manuscript. We would also like to thank the reviewer for pointing it out.
>   * The reviewer highlighted the absence of loss ablation studies. To address this, we have revised the ablation study to assess the performance improvements contributed by each additional loss component. Please refer to **Appendix Section C** in the revised manuscript for detailed information.
>   * The reviewer asks for out-of-distribution results for the baselines. We have now added them to the manuscript. Please refer to **Appendix Section B.1** in the revised manuscript for detailed information.
>   * The reviewer suggested improving Figure 2 with the meaning of the notations used in the figure. Additional meaning of notations and highlighting the individual modules have been now included in Figure 2 of the new manuscript.
>   * In Figure 1, **all three G represents the same dual**. We have now overlaid the graph on the hand to make it clearer. In regards to the adjacency matrix generation, we fixed the ordering of the triangles in the dual graph. The Isomap then outputs the ordered set of vertex locations. We have now mentioned the following process in Figure 1 to make it more clear.
>
> **Follow-Up and Clarifications**
>
> * Strength of the contribution **(W1)**
>   * The reviewer questions the novelty of our work as we mention a lot of references for the polygon reconstruction problem. While it is true that polygon reconstruction is a well-studied problem, ours is the **first work** to **solve a classical computational geometry problem (visibility reconstruction, characterization, and recognition) using a learned representation**, and showcasing **an SDF intermediate representation is much more effective** for the blocks to understand polygon mapping to **combinatorial structures that lack well-behaved local neighborhoods or metric structures**. We also explicitly detail why current polygon reconstruction approaches are not designed for the problem of visibility reconstruction in our related work section. We hope that the reviewer appreciates these novel aspects of our work.
> * Result present in complicated setting **(W1)**
>   * The reviewer mentions that results already exist for polygons with holes, pseudo-polygons, and other different types of polygon types which are more complicated than our problem. These are **theoretical results and specific to restricted polygon types**. Our focus is on developing a general representation learning model that works across all polygons, an area not currently explored. We hope to extend our work to pseudo-polygons and polygons with holes in the future.
> * Datasets **(W1, W2)**:
>   * The reviewer mentions that there are existing datasets and baselines for our work in **W1** and **W2**. We would appreciate it if the reviewer could kindly specify the datasets or baselines they are referring to in **W1** and **W2**. We would be happy to incorporate comparisons with them to further strengthen our evaluation.
>   * The reviewer also mentions increasing the number of vertex locations in the dataset. We recognize the importance of increasing vertex locations. Our exploration was constrained by computational costs. Increasing the number of points requires a finer grid to capture small visibility changes. A Finer grid results in a significant increase in computational demands. In particular, we trained our model on 1 v100 GPU with 30 GB RAM. The SDF diffusion model with a grid of 0.05 resolution (grid size for 25 vertex locations) uses 10 GB of memory during training. Reducing the grid resolution to 0.01(grid size for 50 vertex locations), the memory consumption increases to 40 GB. To address this, we plan to remove the grid structure entirely in future work, overcoming these constraints.

---

> > ### Author Response · Authors · 2024-11-19
> > **Reply to Reviewer hNLw (Part 2/2)**
> >
> > * Lack of related work on loss **(W1, Q4)**
> >   * The reviewer expressed concern about the lack of a discussion on existing loss functions for our problem in related works. Our work is the first to generate polygons from visibility graphs using representation learning, and no prior loss functions exist for this problem. We believed there was no need to present a discussion on the existing loss functions for our problem in the related works section.
> > * Polygon characterization polygon generation **(Q3)**
> >   * Clarification of the polygon characterization process: The process that the reviewer mentions is correct. As stated in the paper (**Section 6.2.2** in the revised manuscript), we address the characterization problem by generating multiple polygons for the same visibility graph G by drawing different samples from a Gaussian distribution for diffusion initialization.
> > * Clarification regarding why F1 scores in Figure 5 captions do not match the F1 scores from Table 1 **(MC5)**
> >   * We would like to point out to the reviewer that the figure captions indicate the F1 score with the visibility of the ground truth polygon shown in Figure **5a**. Table **1** presents the quantitative performance across the entire test dataset.
> >
> > We would like to finally thank the reviewer for their time and efforts in reviewing our work and giving constructive feedback. Please let us know if there are any further questions or clarifications, we would be happy to address them.

---

### Official Review · Reviewer_qWQ2 · 2024-11-08

**Soundness:** 2
**Presentation:** 2
**Contribution:** 3
**Rating:** 6
**Confidence:** 3

**Summary:**

This paper presents a way to infer polygonal information from given combinatorial structure, such as visibility graph or triangulation. To do this, this paper proposes to use a diffusion model. The model takes in a visibility graph as an input, and first generates a SDF, from which they select vertex locations, which satisfy the given visibility graph. To train this model, this paper proposes a loss function for comparing the output polygon's visibility graph with the given one in a differentiable manner. The loss formulation is based on geometric algorithms, such as line-line intersection. One another contribution is in using cancavity-aware dataset for training. By using this approach, it achieves much better performance in reconstructing valid polygons from visibility graphs than the other baseline methods. Also, as this approach can sample multiple polygons from different seeds, it can be used for visibility characterization. Finally, it could give a hint about solving recognition problem.

**Strengths:**

1. The paper was easy to follow, as the target problems were explained clearly.

2. The approach seems to be novel. It cleverly applied generative model for solving this kind of geometric / topological problem. However, I could be wrong as I'm not an expert about this topic.

3. The results clearly show that this approach is better than the other baselines.

**Weaknesses:**

1. There is no comparison with the baselines for the out of distribution data (Table 2), which makes it hard to see if this approach is better than the baselines for such data.

2. As far as I know, the diffusion denoisinig process takes some computational cost, and this paper says that it generated 50 polygons from the given visibility graph and chose the best one among them, which might have led to longer computation time than the other baseline methods. It would be better to have analysis about computational cost.

3. Generally, the figure quality is not very good. It would be better to use figures of better rendering quality. Also, I believe Figure 2 can be further improved (it is too abstract).

**Questions:**

1. Can we use Unsigned Distance Field (UDF) instead of SDF, so that we can cover wider range of polygons (e.g. polygons with holes)?

2. Can we extend this approach to 3D? If we can, it would be really cool. I assume there would be much more difficulty in vertex selection & connecting them to get valid watertight mesh.

3. On line 236, it says that "the generated SDF embedding is then passed through multiple MLP layers to predict the **ordered** vertex locations". I think there are multiple (valid) ordered vertex locations for the same polygon, right? For instance, vertex ordering of (1-2-3-4) would be same as (2-3-4-1). How did yout resolve such ambiguity? Didn't it raise any problem during training?

4. How does the performance change when we increase the number of points?

* minor typo

Line 496: "...threshold X, it is ..."

---

> ### Author Response · Authors · 2024-11-19
> **Reply to Reviewer Reviewer qWQ2**
>
> Thank you for your overall positive evaluation and helpful feedback. The reviewer’s main concerns were about the **baseline results on out-of-distribution data, computational cost, figure 2 explanation, future extensions of our work, increased number of points in the dataset, and addressing ordering ambiguity**. We address each of the reviewer’s concerns as follows-
>
> We would like to first thank the reviewer for their great suggestions which we have now added in the manuscript.
>
> * Ordering ambiguity **(Q3)**, Baseline results on out-of-distribution data **(W1)**, Computational cost comparison **(W2)**, and Figure 2 improvement **(W3)**
>   * We thank the reviewer for the valuable suggestions.
>     * Baseline results on out-of-distribution data **(W1)**: We add the performance metrics of baselines on our out-of-distribution test set. Specifically, we add them to **appendix section B.1** (Tables **3,4,5,6**, and **7**) in the revised manuscript.
>     * Computational cost comparison **(W2)**: An inference time comparison between our approach and the baselines has been included in **appendix section B.3** (Table **9**) of the revised manuscript.
>     * Ordering ambiguity **(Q3)**: In regards to ordering ambiguity, we fixed it to be anticlockwise. We now mention the ordering detail in **Section 5** of the updated manuscript.
> * Figure 2 improvement **(W3)**: Additional meaning of notations and highlighting the individual modules have been now included in **Figure 2** of the new manuscript.
>
> **Clarifications to reviewer’s questions**
>
> * Use of unsigned distance **(Q1)**
>   * The reviewer asks if unsigned distance could be used so as to handle polygons with holes. We would like to clarify that regular SDF is also well-defined for polygons with holes. We are not sure about the benefit of using a different implicit function.
> * Extension to 3D **(Q2)**
>   * The reviewer suggests extending the model to 3D. We agree with this valuable direction of applying the approach to 3D. We focused on the 2D case in our current work to maintain a clear scope. Extending to 3D is an exciting avenue we plan to explore in future work. Thank you for highlighting this potential.
> * Increased number of points **(Q4)**
>   * The reviewer asks about the performance change with the increased number of points. Increasing the number of points requires a finer grid to capture small visibility changes. A finer grid results in a significant increase in computational demands. In particular, we trained our model on 1 v100 GPU with 30 GB RAM. The SDF diffusion model with a grid of 0.05 resolution (grid size for 25 vertex locations) uses 10 GB of memory during training. Reducing the grid resolution to 0.01(grid size for 50 vertex locations), the memory consumption increases to 40 GB. As mentioned in the conclusion and future work section, we plan to remove the grid structure entirely to overcome this limitation.
>
> We would like to finally thank the reviewer for their time and efforts in reviewing our work and giving constructive feedback. Please let us know if there are any further questions or clarifications, we would be happy to address them.

---

> > ### Comment · Reviewer_qWQ2 · 2024-11-27
> >
> > Thank you to the authors for addressing my questions—I greatly appreciate it. The additional experimental results provided in Section B.1 and Section B.3 were particularly helpful.
> >
> > However, I believe there is still room for improvement in the paper's presentation, especially with regard to figure layout. For example, on page 9, a small paragraph is awkwardly placed between Table 2 and Figure 6. Additionally, many figures appear to be vertically compressed to fit the space (e.g., Figures 5, 6, and 7), and the resolution of Figure 7(e) is notably low.
> >
> > While I commend the novelty of the work, which I found compelling, these issues significantly impact the overall readability and presentation. Therefore, I am not convinced that this paper is ready for publication with only minor modifications at this stage and will maintain my current score.

---

### Official Review · Reviewer_kFhu · 2024-11-09

**Soundness:** 3
**Presentation:** 2
**Contribution:** 2
**Rating:** 5
**Confidence:** 3

**Summary:**

The visibility graph of a planar polygon is the graph whose vertices are those of the polygon and two vertices are linked with an edge iff the line segment connecting them lies inside the polygon. The paper presents a deep model to generate polygons from a visibility graph. This model is then used to sample polygons approximately consistent with a given visibility graph. As a side result, this is also applied to deciding whether a given graph is a (valid) visibility graph of some polygon.

The main block of the model is diffusion, based on U-Net. The output of the diffusion is a signed distance function (represented as a matrix) from the polygon boundary. The SDF is finally regressed to a set of polygon vertices.

The dataset for training and testing consists of polygons each with 25 vertices, First, 60k polygons are  randomly generated by an existing heuristic. Then they are augmented by shear and small vertex perturbations to obtain 400k polygons of random. This dataset defines an empirical distribution both on visibility graphs and on polygons consistent with a given vis. graph. For testing, other out-of-distribution polygons are generated, based on 5 base shapes.

The model is compared to several known deep models suitable for the  task (such as transformers or GNN) and a direct approach based on optimiation (Nelder-Mead amoeba algorithm). The model is reported to perform best in all criteria, except sometimes accuracy.

**Strengths:**

To my understanding, this is the first paper to design a deep model specifically for generating polygons from visibility graphs.

**Weaknesses:**

Novelty:

First, I am not sure about novelty. On one hand, this paper seems to be the first to propose a deep generative model for generating polygons from vis. graphs (I think so because no such paper is cited). Though there has been a lot of theoretical results and algorithms on visibility graphs of polygons, these are not deep models.

On the other hand, I am not sure if this alone should guarantee acceptance. The proposed model is composed of well-known blocks and does not bring any really novel ideas - perhaps except using SDF as an intermediate representation of polygons. Moreover, the practical utility of the task is questionable. There are many very different polygons consistent with a single vis. graph and vis. graph is unstable to small polygon perturbations. Indeed, look at Figure 5: the polygons sampled from the vis. graph are all very dissimilar to the GT polygon. I believe that  for the tasks like representing a hand (mentioned in the intro), representation that includes also metric info would be better. I do not see many applications benefiting from such a model.

Experiments:

As no deep models specifically tailored for the task have been published before, all competing models in the experimental comparison seem to be coded by the authors themselves, using known general deep learning models and no details on these models are given (code is promised but not yet available). This is not seen from the text (sorry if I overlooked something) until you look at the references cited with the models. This may introduce positive bias towards the published model, since the authors may have given more care to implementing this model rather than the other methods. E.g, it is mentioned on line 452 that in visibility reconstruction, 50 polygons are sampled from the vis. graph and the best of them is picked. Is this done also in the competing methods?

The polygons in the dataset are all of the same "type", obtained by a single heuristic. It would be more informative to have a richer distribution of polygons.

In visibility recognition (Section 6.5), my understanding is that the way how "non-valid" vis. graphs are generated may sometimes produce valid ones. At least, I do not see why the vis. graph generated using a polygon with a hole should be invalid, because there can be some polygon consitent with it. Please exaplain.

Clarity:

The technical parts of the paper are sometimes unclear and imprecise. The worst is Section 4 on designing the loss function (a key section of the paper): the part on the visibility term is unreadable. Examples:

Line 189, "the dimensionality of the input 25-by-25 matrix is reduced to 512": Since 25*25 = 525, is this worth it? Moreover, one can use only a half of the vis. graph adjacency matrix, since it is symmetric.

Paragraph starting at line 275: I do not see how the visibility loss term is computed, too little detail is given. Moreover, math formulas in this par is inacceptably informal. One cannot write things like max(L_int), because what is L_int? Softmax on the rhs of (6) has not been defined. In (7), i\neq j should be in the sedond sum, P_i was not defined.

Line 282: What function precisely is differentiable?

Eq. (3): one cannot write this, you need a norm instead of just the parentheses (and what is X, exactly?).

Line 515: There is no such thing as concave polygon (or concave set), correct word is "non-convex".

**Questions:**

See Weaknesses.

---

> ### Author Response · Authors · 2024-11-19
> **Reply to Reviewer Reviewer kFhu (Part 1/2)**
>
> Thank you for your overall positive evaluation and helpful feedback. The reviewer’s main concerns were about the **novelty of the work, baselines, evaluation methodology, dataset generation, non-valid experiment, and loss function**. We address each of the reviewer’s concerns as follows-
>
> **Note:** Here **Experiments** represent the comments made by reviewer in their experiments section. **Novelty** represents the comments made in the novelty section of the reviewers while **Clarity** represents the clarity section of the reviewer
>
> * Novelty and Practical Applications **(Novelty W1)**: The reviewer acknowledges that our work is the first approach that attempts to solve the reconstruction and recognition problems by learning a new representation. However, they also question the novelty of our approach since we use well-known blocks in our final architecture. We respect the reviewer's opinion but also note that the choice of this architecture was not obvious to us. We tried many different approaches (some of them documented in baseline comparisons) to arrive at this specific architecture. In terms of practical applications, we acknowledge that at this point our motivation is primarily theoretical. We are planning to explore practical applications such as generating balanced datasets of polygons (to be used, for example, for training motion planners) in our future work.
> * Baselines **(Experiments W1)**
>   * The reviewer is concerned about the positive bias towards the published models. Since there are no available baselines for our approach. To ensure fair evaluation, we **used standard implementations of blocks** (e.g., Transformer Decoder, Attention) available in PyTorch to design the architecture, and Scipy (for Optimization). We also experiment with **varying numbers of layers** and take the best one to make sure the comparison is as fair as possible.
> * Selection Process **(Experiments W1)**
>   * The reviewer is concerned about the selection process (selecting the best one) of the polygon being fair in all the baselines. We would like to clarify that the best selection process is only feasible for diffusion models, as they have the ability to generate multiple instances for the same visibility graph. Hence, a similar selection process was applied to both the vertex diffusion model and our approach. For other baselines, such as Sequence Prediction, Encoder-Decoder, and Optimization, which can only generate a single sample per visibility graph, a best-selection method cannot be applied.
>   * We also experimented with VAEs for our problem with the same generation setup as diffusion (50 samples and selecting the best) and observed 77% accuracy on the visibility reconstruction problem for 15 vertex locations. We did not include this result in the paper since the number of vertices (15) was too low and training for 25-vertex polygons was taking too long. To address the reviewer's comment, we are currently in the process of training it for 25 vertex locations and intend to add it as an additional evaluation baseline when the training and evaluations are complete (3 days).
> * Dataset **(Experiments W2)**
>   * The reviewer mentions that our dataset consists of the same “type” of polygon. We would like to clarify that our dataset includes diverse polygons, quantified using the “link diameter property” of the visibility graph. The process has been detailed in the dataset generation section of the paper. **We do not use any heuristic or have a single type of polygon in the dataset**.

---

> ### Author Response · Authors · 2024-11-19
> **Reply to Reviewer Reviewer kFhu (Part 2/2)**
>
> * The experiment regarding non-visibility graphs **(Experiments W3)**
>   * In order to generate graphs that are not visibility graphs, we used visibility graphs of polygons with holes. The reviewer rightly expressed concern that the fact that a graph is the visibility graph of a polygon with holes, does not mean that it is not a visibility graph (of a simply connected polygon). We struggled with the same issue ourselves since we could not find a way to generate graphs which are guaranteed to be non-visibility graphs. We are also aware that there are construction tricks one can use to turn a polygon with holes into a simply connected polygon (by opening a small tunnel from the hole to the nearest point on the boundary) but those tricks introduce additional vertices and change the visibility structure (the tunnel can hit visibility edges). Furthermore, the visibility structure of polygons with holes has unbounded VC-dimension whereas that of a simply connected polygon is constant (Valtr’98, Isler et al’04). Therefore, using polygons with holes to construct non-visibility graphs seemed like a plausible heuristic. We are very much open to suggestions for other classes and happy to perform additional experiments. For example, we can try random graphs but they have the same issues (potentially a visibility graph) and probably are an easier set to deal with. Thank you again for bringing up this issue! We welcome additional thoughts and suggestions.
> * Loss Function **(Clarity W3 and W4)**
>   * The reviewer mentions that the loss section is not readable as the definitions of terms are not defined. To assist the reviewer in better understanding the loss function, we would like to highlight the specific figures and line numbers where the relevant definitions are provided-
>     * Softmax definition **(Clarity W3)**, the norm for Eqn **3** **(Clarity W5)**: We would also like to thank the reviewer for their specific suggestions on mathematical notations and missed softmax definition which we have now applied to our revised manuscript.
>     * L_int, L_out **(Clarity W3)**
>       * L_int is described in Figure **3** of the manuscript. We kindly direct the reviewer to Lines **288** - **297** for a specific explanation of L_out in the revised manuscript.
>     * X **(Clarity W3)**
>       * We kindly direct the reviewer to Line **264** for the explanation of X in the revised manuscript.
>     * P_i **(Clarity W3)**
>       * We kindly direct the reviewer to Line **314** for an explanation of P_i in the revised manuscript.
>     * max(L_int)  **(Clarity W3)**
>       * We would be very grateful if the reviewer could elaborate on the reasoning behind max(L_int) being considered non-valid. As mentioned in Figure **3**, L_int is calculated between each vertex pair. max(L_int) identifies the pair with the maximum value of the Int function.
>     * What function precisely is differentiable **(Clarity W4)**: The reviewer also mentions not being able to understand which function is differentiable in visibility loss. We would like to specifically mention here that our work provides a way to extract the visibility graph from the polygon during training in a differentiable manner. Since visibility graph computation involves non-differentiable operations like intersection and inside-outside tests. We propose a differentiable method to compute the visibility graph, enabling its use as guidance in the loss function.
>
> We would like to finally thank the reviewer for their time and efforts in reviewing our work and giving constructive feedback. Please let us know if there are any further questions or clarifications, we would be happy to address them.
>
> **References**
> * [Valtr’98] Valtr, Pavel. "Guarding galleries where no point sees a small area." Israel Journal of Mathematics 104 (1998): 1-16.
> * [Isler et al’04] Isler, Volkan, et al. "VC-dimension of exterior visibility." IEEE Transactions on pattern analysis and machine intelligence 26.5 (2004): 667-671.

---

> ### Author Response · Authors · 2024-11-22
> **Additional Quantitative Results with VAE**
>
> Selection Process **(Experiments W1)** :  Addressing the reviewer's concern about the fairness of the selection process (choosing the best one) across all baselines. We trained a **VAE** network to generate vertex locations of polygon conditioned on the visibility graph and perform similar comparisons on in-distribution test set. The selection process (generating 50 samples and selecting the best one) for the VAE is same as **VisDiff**, ensuring fair evaluation. The results are as follows:
>
> | Approach | Accuracy | Precision | Recall | F-1 Score|
> |----------|----------|----------|----------|----------|
> | VisDiff   | **0.85**   |  **0.83**   | **0.77** |  **0.80** |
> | VAE   | 0.66   | 0.54   | 0.70 | 0.60 |

---

### Official Review · Reviewer_xawV · 2024-11-10

**Soundness:** 3
**Presentation:** 2
**Contribution:** 2
**Rating:** 5
**Confidence:** 2

**Summary:**

The paper presents a diffusion model for 2D polygonal shapes represented in terms of their visibility graph (a skeleton-like structure obtained by taking the dual of a triangulation of the shape)
The diffusion model generates the signed-distance-function (SDF) of the polygonal shape constrained by the structure of the visibility graph, then the actual vertices of the polygon (at the zero level set of the SDF) are extracted by a second module, where an encoder with Spatial Transformer Cross Attention encodes the SDF into a latent space, and then several MLPs decode the vertices.

**Strengths:**

Good performance with respect to the baselines on the generated dataset

**Weaknesses:**

1) the problem is a bit contrived, in the sense that the way it is evaluated the focus is not in shape analysis, but on extracting something from the chosen skeletal representation. This results in no analysis on the underlying problem, thus not using any of the several datasets that have been used by the community for several years, and comparisons to approaches that can only be described as baselines for the problem actually analyzed. This severely limits the readership of this paper.

Further, I had to read the papers several times because key elements (such as the visibility graph)  do not have proper definitions, but are defined inline in the text.

**Questions:**

You have the diffusion model generate the SDF and then the prediction model re-encode the SDF to generate the vertices. It would be interesting to see how much of the performance is due to the diffusion, and how much is re-aligned by the prediction module. In other words, how good are the SDFs before the prediction module?

EDIT: I was thinking how to measure this, as it always hard with generative models where you do not have a ground truth. One way to al least understand whether the generated SDF fits the final shape is to compare the generated SDF with the actual SDF of the shape generated by the prediction module. This will not tell us how good the SDF was, but at least will gives us an idea of how much leeway the prediction module has.

---

> ### Author Response · Authors · 2024-11-19
> **Response to Reviewer xawV**
>
> Thank you for your overall positive evaluation and helpful feedback. The reviewer’s main concerns were about the **problem formulation, evaluation approach, dataset, baselines, SDF evaluation, and definitions**. We address each of the reviewer’s concerns as follows:
>
> We would like to first thank the reviewer for their great suggestions which we have now added in the manuscript.
> * Definitions **(W2)**
>   * The reviewer is concerned about the visibility graph definitions being defined inline. We have now added the definitions of the terms in the appendix. We specifically ask the reviewer to refer to **Appendix A** in the new revised manuscript for more details.
> * SDF Evaluation **(Q1)**
>   * The reviewer is interested in the SDF diffusion model performance. Hence, we include an evaluation table measuring the L-2 distance between the ground truth and predicted SDF. Specifically, we would like to refer to **Appendix B.2** in the new revised manuscript for more details.
>
> **Clarifications and Follow-up**
> * Problem Formulation **(W1)**
>   * The reviewer mentions the problem discussed in the paper is contrived. Visibility graph reconstruction, recognition, and characterization are well-known problems in computational geometry. The existing literature on visibility graphs is also discussed in our related works. Hence, we respectfully disagree with the reviewer that our problem is contrived.
> * Evaluation Approach **(W1)**
>   * The reviewer is concerned about the evaluation metrics being no analysis of the problem discussed in the paper. To clarify, we would like to first reiterate our general formulation of the problem. The reconstruction problem is formulated as follows: “Given a valid graph G, generate a polygon P such that G(P)=G”. The formulation defines generating a polygon P which follows the given visibility graph G. Hence, a direct quantitative evaluation metric of our formulation is by calculating the visibility graph of the generated polygon P and comparing it with the input visibility graph G (Table 1 shows performance on the basis). Therefore we respectfully disagree with the reviewer about the evaluation metric not being relevant to the problem being discussed in the paper.
> * Dataset and Baselines **(W1)**
>   * The reviewer is concerned about not using datasets and baselines that have been used in the community for several years. We would appreciate it if the reviewer could kindly specify the datasets or baselines they are referring to in **W1**. We would be happy to include these comparisons and demonstrate our method's performance in those contexts.
>
> We would like to finally thank the reviewer for their time and efforts in reviewing our work and giving constructive feedback. Please let us know if there are any further questions or clarifications, we would be happy to address them.

---

### Author Response · Authors · 2024-11-19
**Rebuttal Revision 1 Summary**

We would like to thank the area chairs and the reviewers for their time and expertise. We sincerely appreciate the reviewer’s insights, thoughtful efforts, and valuable comments. We were able to incorporate most of their suggestions in the revised manuscript, which has significantly improved the presentation of our work. Below, we first give an overall summary of the revisions. Afterward, we describe how we address individual reviewer’s comments and provide additional clarifications. Additionally, we have included a few follow-up questions for the reviewers, which we believe will help us better address their concerns and refine our work further.

**Summary of Revisions**
* **SDF Evaluation (Appendix B.2)**:We compare the L2 Loss between the ground truth SDF and the predicted SDF from the diffusion model. (requested by reviewer [xawV](https://openreview.net/forum?id=rn8r7GqJm6&noteId=vLnwdnRbUV))
* **Loss function (Section 4)** - (requested by reviewer [xawV](https://openreview.net/forum?id=rn8r7GqJm6&noteId=vLnwdnRbUV))
  * Added norm operator for the MSE Loss (Eq 3)
  * Added i \neq j in the second summation (Eq 7)
  * Added softmax definition (Eq 6)
* **Out-of-distribution data baseline evaluation (Section B.1)**: An evaluation table of baseline performance on the out-of-distribution test set is now included for more comprehensive analysis. (requested by reviewers [qWQ2](https://openreview.net/forum?id=rn8r7GqJm6&noteId=A5gmR2Ake6) and [hNLw](https://openreview.net/forum?id=rn8r7GqJm6&noteId=pTU26U0omM))
* **Computational cost comparison (Section B.3)**: We compare the inference time with the baselines for a computational cost comparison. (requested by reviewer [qWQ2](https://openreview.net/forum?id=rn8r7GqJm6&noteId=A5gmR2Ake6))
* **Figure 2 improvement**: Figure 2 captions were modified to include the meaning of the notations used in the diagram and also highlights the process of polygon generation. Additionally, all modules are also now marked in the figure for a better understanding of the architecture. (requested by reviewers [qWQ2](https://openreview.net/forum?id=rn8r7GqJm6&noteId=A5gmR2Ake6) and  [hNLw](https://openreview.net/forum?id=rn8r7GqJm6&noteId=pTU26U0omM))
* **Ordering Ambiguity(Section 5)**: We missed mentioning an important detail about fixing the ordering of the polygon anticlockwise. We apologize for the oversight and have now mentioned the ordering detail in the dataset generation section (Section 5). (requested by reviewer [qWQ2](https://openreview.net/forum?id=rn8r7GqJm6&noteId=A5gmR2Ake6))
* **Loss function ablation Study (Appendix C)**: To further demonstrate the effectiveness of our loss components. We add an analysis of performance change with different loss components in the ablation study section (Appendix C). (requested by reviewer [hNLw](https://openreview.net/forum?id=rn8r7GqJm6&noteId=pTU26U0omM))
* **Title change and conclusion change**: We now mention loss function and visibility recognition as contributions in the conclusion. Additionally, we also include characterization in the title. (requested by reviewer [hNLw](https://openreview.net/forum?id=rn8r7GqJm6&noteId=pTU26U0omM))
* **Additional reference added in the related work section**: A reviewer mentioned a reference that they think is related to our work. We now add the mentioned reference and a comparison of how our work is different from the problem discussed in the reference. (requested by the reviewer [k2Tx](https://openreview.net/forum?id=rn8r7GqJm6&noteId=MY9X3bVoed))
* **Additional Definitions (Appendix A)**: Addition of definitions of visibility graphs and simple polygons in the appendix for a better understanding of the terminology.  (requested by the reviewer [xawV](https://openreview.net/forum?id=rn8r7GqJm6&noteId=vLnwdnRbUV))
* **Figure 1 clarity**: We include the caption with the process of adjacency matrix generation and also overlay G on the hand to increase clarity about all the G representing the same dual. (requested by the reviewers [xawV](https://openreview.net/forum?id=rn8r7GqJm6&noteId=vLnwdnRbUV) and [k2Tx](https://openreview.net/forum?id=rn8r7GqJm6&noteId=MY9X3bVoed))

**Note:** In individual comments, **Q** represents the questions section of the reviewer, and **W** represents the weakness section of the reviewer.

---

> ### Author Response · Authors · 2024-11-26
> **Rebuttal Revision 2**
>
> * **VAE Baseline**: We have added VAE results as an additional baseline to further demonstrate the effectiveness of VisDiff on the problems discussed in our work. Table 1 now includes VAE quantitative results, and Appendix F has been added to include qualitative and out-of-distribution quantitative results for VAE. (Requested by reviewer [kFhu](https://openreview.net/forum?id=rn8r7GqJm6&noteId=tb9wIP7WTF))

---

### Meta-Review · Area_Chair_HF3q · 2024-12-22

**Metareview:**

The paper studies the problem of reconstructing a polygon from its visibility graph. The main contribution of the paper is a diffusion based approach for the problem, including the development of a loss function and curated dataset for training.

The reviewers found the application of ML techniques to this challenging problem to be a well-motivated direction. The reviewers noted several significant weaknesses. First, the manuscript will need a significant revision to clarify the paper's main contributions and provide a more detailed comparison with prior work. Second, the experimental methodology and the experimental evaluation will need to be significantly strengthened. The authors revised the manuscript based on the reviewers' feedback, but the reviewers noted that the changes were only local and did not address their main concerns. There was consensus among the reviewers that participated in the discussion that the paper will need to be significantly revised and strengthened before it can be accepted.

**Additional Comments On Reviewer Discussion:**

The author response and the author-reviewer discussion sufficiently addressed the reviewers' concerns regarding the novelty of the method. Following the discussion, the reviewers that participated in the discussion noted that their remaining main concerns noted above still remained.

---

### Decision · Program_Chairs · 2025-01-22

Reject